# Biomass burning and marine aerosol processing over the southeast Atlantic Ocean: A TEM single particle analysis

Caroline Dang[1,2], Michal Segal-Rozenhaimer[3,4], Haochi Che[3], Lu Zhang[3], Paola Formenti[5], Jonathan Taylor[6], Amie Dobracki[7], Sara Purdue[7], Pui-Shan Wong[8], Athanasios Nenes[9,10], Arthur Sedlacek III[11], Hugh Coe[6], Jens Redemann[12], Paquita Zuidema[7], Steven Howell[13], James Haywood[14, 15]

*Correspondence to: Caroline Dang (CarolineVanDang@gmail.com) and Michal Segal-Rozenhaimer (msegalro@tauex.tau.ac.il)*

[1]NASA Ames Research Center, Moffett Field, CA, 94035, USA
[2]Oak Ridge Associated Universities, Oak Ridge, TN, 37831, USA,
[3]Department of Geophysics, Porter School, Tel Aviv University, Tel Aviv, 69978, Israel
[4]Bay Area Environmental Research Institute, NASA Ames Research Center, Moffett Field, CA, USA
[5] Université de Paris Cité and Univ Paris Est Creteil, CNRS, LISA, F-75013 Paris, France
[6]Department of Earth and Environmental Sciences, University of Manchester, Manchester, UK
[7]Rosenstiel School, University of Miami, Miami, FL, USA
[8]Mount Allison University, Sackville, New Brunswick, CA
[9]Laboratory of Atmospheric Processes and their Impacts, School of Architecture, Civil & Environmental Engineering, École Polytechnique Fédérale de Lausanne, Lausanne 1015, Switzerland
[10]Center for Studies of Air Quality and Climate Change, Institute of Chemical Engineering Sciences, Foundation for Research and Technology Hellas, Patras 26504, Greece
[11]Brookhaven National Laboratory, Brookhaven, NY, USA
[12]School of Meteorology, University of Oklahoma, Norman, OK, USA
[13]Department of Oceanography, University of Hawai`i at Mānoa, HI, USA
[14]College of Engineering, Mathematics and Physical Science, University of Exeter, Exeter, UK
[15]Met Office, Exeter, EX1 3PB, UK

## Abstract

This study characterizes single particle aerosol composition from filters collected during the ObseRvations of Aerosols above CLouds and their intEractionS (ORACLES) and CLoud–Aerosol–Radiation Interaction and Forcing: Year 2017 (CLARIFY-2017) campaigns. In particular the study describes aged biomass burning aerosol (BBA), its interaction with the marine boundary layer and the influence of biomass burning (BB) air on marine aerosol. The study finds evidence of BBA influenced by marine boundary layer processing as well as sea salt influenced by BB air. Secondary chloride aerosols were observed in clean marine air as well as in BB-influenced air in the free troposphere. Higher volatility organic aerosol appears to be associated with increased age of biomass burning plumes, and photolysis or oxidation may be a mechanism for the apparent increased volatility. Aqueous processing and interaction with the marine boundary layer air may be a mechanism for the presence of sodium on many aged potassium salts. By number, biomass burning potassium salts and modified sea salts are the most observed particles on filter samples. The most commonly observed BC coatings are inorganic salts. These results suggest that atmospheric processing such as photolysis, oxidation and cloud processing are key drivers in the elemental composition and morphology of aged BBA. Fresh BBA inorganic salt content, as it has an important role in the particles' ability to uptake water, may be a key driver in how aqueous processing and atmospheric aging proceeds.

# 1. Introduction

With Africa producing almost a third of the Earth's biomass burning aerosol (BBA) (Roberts et al., 2009), two aircraft campaigns, ObseRvations of Aerosols above CLouds and their intEractionS (ORACLES) and CLoud–Aerosol–Radiation Interaction and Forcing: Year 2017 (CLARIFY-2017) were focused on understanding African biomass burning aerosol interaction with clouds and radiation in the southeast Atlantic (Haywood et al., 2021; Redemann et al., 2021). The CLARIFY campaign was based on Ascension Island in 2017 and sampled primarily in that vicinity,

and ORACLES was based in São Tomé in 2017 and 2018 and generally sampled closer to Africa than CLARIFY. CLARIFY findings detail a complex vertical structure in aerosol with a temperature structure inhibiting mixing between layers (Haywood et al. 2021). Over Africa, mixing is inhibited by stable layers at the top of the continental boundary layer (CBL) (Garstang et al., 1996), and over the southeast Atlantic the BBA in the residual CBL moves over the marine boundary layer (MBL) as the air is transported west (Haywood et al. 2021). However, BBA aerosol

is more often affected by the MBL than previously accounted for, reaching the MBL through pathways that are not fully articulated (Zuidema et al., 2018), with entrainment processes through the clouds potentially altering aerosol properties further. An example of this is the low single scattering albedo in the boundary layer compared to the free troposphere (Zuidema et al., 2018; Pistone et al., 2019). Both campaigns report that a more detailed aerosol process-level understanding including the properties of black carbon, organic carbon and inorganic compounds and how they

vary as a function of mixing state and altitude is needed, as is knowledge of properties of the aerosol as they age from emission to deposition and the degree of mixing of BBA into the MBL (Haywood et al. 2021; Redemann et al. 2021).

      While in-situ instruments provide data over large temporal and spatial scales, the instruments which analyzed chemical composition in the ORACLES and CLARIFY campaigns analyzed bulk aerosol; detailed off-line single particle

analysis can offer valuable information to complement these online measurements. The principal in-situ instrument used in these campaigns to determine aerosol chemical composition is the Aerosol Mass Spectrometer (AMS). The AMS can detect organic and non-refractory inorganic mass at high time resolution. There are limitations on the size range of aerosols detected depending on the inlet system employed, with no detection above one micron and a decreasing efficiency above 700 nm. Salts do not vaporize easily and tend to recombine with oppositely charged ions

and make quantification of salts in the mass spectra difficult, if not impossible (Nash et al., 2006). The mixing state of organic and inorganic constituents can only be determined with offline analysis of collected samples rather than in-situ bulk aerosol measurements. Transmission electron microscopy (TEM) coupled with energy dispersive X-ray (EDX) is suited to understanding physical and chemical properties of individual particles including shape, elemental composition, mixing state, volatility and viscosity, and it is particularly useful for complex aerosol which have been

processed  (Signorell and Reid, 2011; Reid et al., 2018; Li et al., 2003). Therefore TEM-EDX is a useful method for understanding processes affecting aged BB aerosol as well as marine salts which are pervasive over the ocean.

      Previous work of African BBA from the Southern African Regional Science Initiative (SAFARI-2000) showed that the aerosols were primarily composed of black carbon, potassium salts, and organic/sulfur (Liu et al. 2000; Pósfai et

al. 2003; Li et al. 2003).  The SAFARI campaign was mostly focused on BBA that were less aged than particles in CLARIFY and ORACLES.  SAFARI results showed that KCl particles occur in young smoke more often while $K_2SO_4$ and $KNO_3$ particles occur more in aged biomass burning aerosol (Li et al. 2003).  This is due to gas phase oxidation of $NO_x$ and $SO_2$ and the displacement of HCl by the stronger acids $HNO_3$ and $H_2SO_4$ during plume transport. The authors theorized that aging caused sulfate to accumulate on organic and soot particles due to the large amount of

internally-mixed soot/sulfate and organic/sulfate in haze (Pósfai et al. 2003).  Based on the location and composition of the particles, Pósfai et al. (2003) concluded that organic and soot particles were the main CCN constituents of BBA. They determined that organic particles with inorganic inclusions likely contribute to the high cloud nucleating capability of biomass burning particles, and Semeniuk et al. (2007), using environmental TEM, found that the inorganic phases of SAFARI particles took up water while soot and tar balls did not; therefore they determined that

the inorganic content of mixed organic-inorganic particles determined the hygroscopic properties of BBA. SAFARI-2000 samples taken in stratus clouds that capped the boundary layer, distinct from the BB haze layer in the FT, and were dominated by sea salt particles (Pósfai et al. 2003). CLARIFY and ORACLES online observations also show an aerosol population dominated by coated black carbon (BC), organic, and sulfates, consistent with the SAFARI TEM findings of BBA. CLARIFY noted a thick inorganic or organic coating around BC (Taylor et al. 2020), while

ORACLES noted a less thick coating around BC as well as a decreasing amount of coating with plume age (Sedlacek III et al., in prep).  ORACLES AMS data also noted a decrease in organic with plume age (Dobracki et al., 2022). As the single particle soot photometer, used to detect coatings on BC, does not differentiate between organic and inorganic material, TEM can help elucidate the type and source of coating on BC.

Sea salt aerosol, generated through a bubble bursting process on the sea surface (Lewis and Schwartz, 2004), have implications for radiative effects (Murphy et al., 1998) and cloud condensation nuclei (CCN) activity (King et al., 2012).  Sea salt aerosols are modified when they react with sulfate, nitrate, and organic acids, resulting in a Na-rich and Cl-depleted aerosol and emission of gaseous HCl (Gard et al., 1998).  There have been studies on the interaction of urban and anthropogenic sources with marine aerosol (Adachi and Buseck 2015), but single particle studies of sea

salt aerosol and variations due to mixing with BB air are scarce.  Coastal areas near urban sites show sea salt particles being modified by anthropogenic sources.  Adachi and Buseck found that sea salt particles were modified by $H_2SO_4$ and $HNO_3$  by acid displacement of Cl (2015) and sea salt particles have also been shown to be Cl-depleted by organic acid displacement (Laskin et al., 2012; Kerminen et al., 1998).  Pósfai et al. (1995) performed TEM analysis of marine aerosol as part of the Atlantic Stratocumulus Transition Experiment/ marine Aerosol and Gas Exchange

(ASTEX/MAGE) campaign and found that polluted continental air affected sea salt aerosol processing, heterogeneity, and mixing with sulfates and nitrates.

With both biomass burning salts and marine salts being major contributors to aerosol in the southeast Atlantic region, a technique that can detect salts is important to accurately represent the aerosol in the region.  Further, the plumes

sampled during the CLARIFY and ORACLES campaigns are aged up to 15 and 7 days, respectively, according to back trajectories initialized at filter sampling times and locations.  This is different from previous campaigns such as

SAFARI-2000, which was deployed closer to the burning source, and so TEM results can provide information on processing of aged (2-15 days from emission) BBA. This paper will describe the single particle analysis in context of the ancillary data including AMS measurements, back trajectories, cloud processing, time from source and time in the MBL. Our main questions are as follows: (1) what are the dominant aerosols in the region and do CLARIFY and ORACLES aerosol differ from each other based on differences in BB plume age, (2) what are the differences observed between MBL and FT aerosol, and (3) what are the proposed processes which have acted on the aerosol. We proceed with a description of filter sampling and analysis methods, describe the region's aerosol types during the two campaigns while comparing and contrasting between the two. Then, we compare aerosol composition and state in the MBL and FT, and discuss possible processing during transport.

**2.0 Method**

**2.1 Filter Sampling**

Aerosol sampling was performed with the NASA Ames Research Center (ARC) aerosol filter system (AFS), installed on the P3, and the filter system operated on the UK Bae-146 aircraft operated by the Facility of Airborne Atmospheric Measurement (FAAM). Lacey carbon TEM grids (Ted Pella, Inc, #01881) were attached to 400 nm hole size polycarbonate nucleopore (WhatmanTM WHA10417112) filters. The Bae-146 has been used for filter analysis for single particle analysis (Chou et al., 2008), as well as bulk analysis (Sanchez-Marroquin et al., 2019; Hand et al., 2010; Andreae et al., 2000). The AFS was composed of a filter holder manifold with five separate filters, connected to the aerosol in situ suite inlet during ORACLES 2017 and 2018. A vacuum pump connected to a flow meter to maintain flow of 30 liters per minute was used for sampling, with five manually controlled valves that were used to switch the sampling to filter holders. The filter manifold was pre-loaded before each flight with filters. Samples for both campaigns were deposited on TEM grids at the locations, sampling times, and total flow volumes listed in Table S1 of the Supplementary Material.

After sampling, the ORACLES 2017 and CLARIFY 2017 filters were sealed in polycarbonate filter holders and wrapped in Parafilm® and aluminum foil and transported together with ice packs in a cooler and placed in a designated freezer immediately at the University of Manchester. The ORACLES 2018 filters were sealed in the same manner and transported with ice packs and stored in a designated freezer at Tel Aviv University. A preliminary set of TEM analysis was conducted on the ORACLES and CLARIFY 2017 filters at the University of Manchester, and then sealed and transported together with ice packs in a cooler for analysis at Tel Aviv University. Care was taken to maintain similar handling, storage, transport, and analysis of all filters in both campaigns. All data included in this study are from the Tel Aviv University analysis.

Size segregation was not performed during particle sampling. Most observed particles are in the submicron range. It is possible that morphologies or compositions were altered during collection, as in other aerosol TEM studies. For

example compositions of hydrate sulfates have been suggested to change in the TEM chamber or during processing (Buseck and Pósfai, 1999),with acidic particles containing more water spreading more on a TEM grid than neutral species. Andreae et al. (1986) suggest that $CaSO_4$ observed on filters without sea salt ions in the marine atmosphere could be from breakup up sea salt particles containing a gypsum crystallite. A sodium chloride core and magnesium chloride coating has been suggested to be due to efflorescence of a particle after collection (Ault et al., 2013). Posfai et al. suggest that an interesting sulphate crystalline rod morphology may be due to water loss within the TEM chamber. Generally, the particles we observed were separated from other particles on the filter and so agglomeration and aggregation did not influence organic mixing with adjacent particles. Samples were collected, on average, for approximately ten minutes and in dry conditions, which may help to limit any chemical reactions the particles are subject to as the aircraft passes into new air masses.

**2.2 Transmission Electron Microscopy – Energy Dispersive X-Ray Analysis (TEM-EDX)**

A JEOL™ JEM-2010F FEG-TEM with a ThermoNoran™ energy dispersive X-ray detector (EDX) was used at Tel Aviv University's Exact Sciences' electron microscopy laboratory to analyze 14 filters from CLARIFY (2017) and 16 filters from the ORACLES (2017, 2018) campaigns. TEM analysis was performed at 200 KeV accelerating voltage, a take-off angle of 15.9 degrees for X-ray emission from the sample, with an electron beam dwell time of no more than 30 seconds and spot size 3. The filter was scanned visually and representative particles near the center of the TEM grid were analyzed. TEM analysis has been known to underrepresent particles under 300 nm (Posfai et al. 2003). EDX spectra was collected for each particle and elemental weight percentage and atomic percentage were found per particle and normalized to 100% using NSS software with Cliff-Lorimer Absorbance correction method. C and O are considered semiquantitative due to the contribution from the Formvar film of C and O from the TEM grid. Ratios of elements such as Na/S and Na/Cl were found by obtaining either the weight or atomic values for individual particles, finding the ratio of interest, and averaging the ratio per filter.

**2.3 Back-trajectory analysis**

Back trajectories of each sample were generated using the Hybrid Single-Particle Lagrangian Integrated Trajectory (HYSPLIT) model (Stein et al., 2015), with the time step set to one hour. Filter sampling lasted up to approximately ten minutes per filter, and back trajectories were calculated as an ensemble of each minute of filter sampling time. To improve the accuracy of the trajectory, we used the hourly high-resolution ERA5 reanalysis data (fifth-generation atmospheric reanalysis data) to drive the calculation. This method captures large-scale movements of air masses, and has some inherent uncertainty; for example, it cannot capture entrainment. The ERA5 data is on $0.25° \times 0.25°$ horizontal resolution and includes 37 pressure levels. We colocated the cloud liquid water content of ERA5 to the coordinates of trajectories, with a threshold of 0.001 g/kg to detect clouds on the trajectory. Two collocations were performed: one with the 4-D coordinates of the trajectory (time, longitude, latitude, altitude) and another one with 3-D coordinates (time, longitude, latitude). Thus, the cloud liquid content points and profiles at the trajectory are

provided, and the mean time of trajectory inside the cloud (cloud liquid water content >0.001 g/kg), and under clear sky (no cloud liquid water above the trajectory) are calculated accordingly. For each sample, we calculated the back trajectories for 1, 2, 3, 5, 7 and 10 days, and the in-cloud and clear sky time correspondingly.

To determine fire locations, fire radiative power (FRP) data was measured by the spinning enhanced visible and infrared imager (SEVIRI) from the geostationary satellite Meteosat-8. The FRP is produced with a 15-min repeat cycle for pixels which contain active burning (Roberts et al., 2005); hourly data was used to match the time step of the trajectory. The age of the BB aerosol is then estimated as the time in days when the trajectory first intercepts the FRP points, similar to the method used by Vakkari et al. (2018). The first FRP interception point with the backtrajectory was chosen, representing the minimum aerosol age, as older aerosol may be more diluted in the plume. The BBA 7-day backtrajectory overlaid with MODIS landcover classifications is included in the Supplementary Material Figure S1.

**2.4 Aerosol Mass Spectrometer, Single Particle Soot Photometer and Cloud Droplet Probe**

The non-refractive chemical composition for submicron particles was measured using two Aerodyne Time-of-Flight Aerosol Mass Spectrometer (ToF-AMS, Aerodyne Research Inc.), a compact version (C-ToF-AMS) used in CLARIFY (Wu et al., 2020) and a high-resolution (HR-ToF-AMS) used during ORACLES (Dobracki et al., 2022; Redemann et al. 2021). The mass concentrations of organic, sulfate, nitrate, and ammonium were provided. Organic aerosol fractions including f43 and f44 were also derived from the mass spectra obtained during both campaigns. f43 is the fraction of the measured organic mass at m/z 43 relative to the total organic aerosol (OA) mass concentration and is indicative of non-acid oxygenates (Ng et al., 2011) common of fragments of aldehydes, ketones, and acid functionalities. Likewise f44 is the fraction of the measured organic mass present at mz44 relative to the total OA mass concentration. The m/z 44 mass is due to acids or esters (Ng et al., 2011). Since these compound classes are commonly associated with low volatility organic fractions, a high f44 has been associated with low volatility aerosol (Aiken et al., 2008).

The mass concentration of the refractory black carbon (rBC) of particles ranging from 80 – 650 nm was obtained in both campaigns by single-particle soot photometers (SP2, Droplet Measurement Technologies, Boulder, CO) using laser-induced incandescence. Detailed information on SP2 measurements can be found in Taylor et al. (2020). Cloud droplet probes (CDP) (Droplet Measurement Technologies, Boulder, CO) were used in both campaigns to measure cloud droplet number concentration.

**3.0 Results**

**3.1 Overview of observations**

Table 1 shows the conditions in which the filters were collected along with ancillary indicators including latitude and longitude, collection above or below cloud, and AMS data including organic, $SO_4$ $NO_3$, and $NH_4$ mass and fraction of PM1 as well as BC mass and number concentration. Time in-cloud in the 24 hours prior to filter collection and time from fire provide additional context for the sampled aerosols. The gaps in the AMS values are due to quality assurance checks which determined that the data for specific filters are unreliable. In the "time from fire" column, if back trajectory analysis did not show interception with fire but rather a marine source, "marine" is noted in the column. There are more samples taken above-cloud, and generally, BC mass values are higher in above-cloud samples. 6/14 CLARIFY filters and 3/16 ORACLES filters were sampled in the MBL, with the remainder sampled in the FT. The ORACLES 2017 and CLARIFY 2017 filters were sampled from mid-August to early September, while the ORACLES 2018 filters were collected late September through October. The ORACLES samples, in general, represents aged BBA and CLARIFY samples represent extremely aged BBA.

Figure 1 indicates the location of filter sampling as well as back-trajectories including altitude per filter. As shown by the back trajectories, the filter samples covered different BB sources such as savanna, forest and grasses, with fires focused around central and southern Africa. Detailed information on ORACLES flight and sampling conditions, per flight can be found in Redemann et al (2021), which provides ancillary data such as CO which will show whether a plume was sampled, with models (Redemann et al. 2021) showing that plumes are often above-cloud.

## 3.2 Aerosol classifications

The TEM filters showed a heterogeneous aerosol population with variations in mixing for organic, NaCl salts, potassium salts, and black carbon. ~30-70 particles on each filter were analyzed to determine composition and particle type. The main particle types including potassium salts, sea salt, black carbon and organic aerosol will be described along with the main findings in the following sections.

### 3.2.1 Organic Aerosol

The AMS data corresponding to filter collection times show that 35% to 70% of CLARIFY and 18% to 68% of ORACLES PM1 is organic, by mass; therefore in-situ data indicates that a substantial amount of PM1 aerosol is organic in both campaigns. While TEM results show organic aerosol for both CLARIFY and ORACLES filters, there is significantly more organic aerosol present on the ORACLES filters. We hypothesize that this is due to differences in the volatility and viscosity of the organic material. Figure 2, left panel, shows a comparison of the fraction of particles, by number, with organic on each filter and the AMS organic fraction, by mass, for the corresponding filter. The majority of CLARIFY filters do not have any particles with organic material, while the majority of ORACLES filters have some particles which contain organic material. This extends to any organic coatings as well; ORACLES organic coatings are largely more thick than organic coatings present on CLARIFY filters. As AMS data shows a significant amount of organic aerosol present in both campaigns (Wu et al. 2020; Redemann et al. 2021), the

differences in visible organic material on filters can be attributed to loss of volatile organics in the TEM chamber. It is known that volatile species will be lost from particles in a TEM chamber (Pósfai et al. 2003; Hudson et al. 2004) and preferential loss of organic would indicate a comparatively volatile material.

For context, Figure 2, right panel, shows the f43 vs f44 space for the entire ORACLES and CLARIFY campaigns,
with filter data overlaid and marked by filter collection below cloud as well as the CO values marked in the colorbar to denote whether the sample is from a BB plume. ORACLES filters (triangles) are 2-7 days and CLARIFY (squares) are 4-15 days aged. A CO cutoff value of over 120 ppbv is used to denote BB-influenced air, based on overall campaign data and Figure 17 in Haywood et al. (2021), which shows the Ascension Island CO frequency distribution and that 120 is at the upper end of the Gaussian distribution of the clean air data. Low volatility oxygenated organic
aerosol will typically have a lower f43 and higher f44 than semi volatile oxygenated organic aerosol (SV-OOA) (Ng et al., 2010, 2011). Most of the variation in filters sampled is in the ORACLES points with higher f44 than the CLARIFY data. As f44 is an indicator of low OOA fraction but not high volatility fraction, the higher ORACLES points with regard to f44 is consistent with TEM findings of lower volatility organic on ORACLES filters. The f43 spread is similar to differences in instrument baselines and therefore should not be over interpreted.


TEM has been used to differentiate high and low contact angle particles where viscosity and volatility of each particle can be qualitatively determined from the particle image. While factors such as surface tension and adhesion forces influence particle shape, viscosity and volatility can still be qualitatively measured on a comparative basis by using electron microscopy images (Reid et al. 2018). Figure 3, top panel, shows a progression from left to right of
increasingly volatile organic as imaged by the TEM. The presence of more rounded, viscous organic (Figure 3, top panel, left image) in ORACLES samples compared to CLARIFY's low contact angle organic (Figure 3, top panel, right image) on the filters is also indicative of relatively higher volatility of organic in CLARIFY filters. More than 80% of ORACLES organic has a rounded morphology, as shown in the left and center panels of Figure 3, top panel.

Figure 3, bottom panel, shows the reduction in Org/BC and Org44/BC mass ratios, based on AMS measurements, for both CLARIFY and ORACLES filters as age from biomass burning source is increased. Filters where backtrajectories did not indicate a BB source are included in the "marine" category. It appears that increased age reduces the organic to BC fraction, similar to the findings of Dobracki et al. (2022) which found organic aerosol to black carbon mass ratios decreasing from 14 to 10 as the aerosol aged over the Atlantic. UV exposure can work to break down oligomers
and low-volatility components in organic (Wong et al., 2015; Lignell et al., 2014) and may account for lower amounts and/or higher volatility of organic present on CLARIFY filters. Photooxidation can also lead to fragmentation of organic chains and oxidation has been observed to change BBOA volatility in laboratory studies (Jahn et al., 2021) as well as physical properties (Jahl et al., 2021). Our results of less organic present for aged samples are consistent with Dobracki et al. (2022) and Sedlacek III et al.'s (in prep) findings of loss of organic and organic coating with age,
although TEM results are caveated by preferential loss of volatile organic.

Tar balls are a type of round organic aerosol unique to biomass aerosol and as of now, the only way to identify tar balls has been through microscopy. Tar balls are estimated to contribute up to ~30% of BB aerosol mass (Sedlacek III et al., 2018). They are highly spherical, high viscosity, and largely resistant to electron beam damage. SAFARI found a considerable number of tar balls (Pósfai et al., 2003) as well as the Biomass Burning Observation Project (BBOP) (Sedlacek III et al., 2018). Adachi et al. (2019) observed tarball formation, likely from primary organic particles, within three hours of emission, with the processing of tar balls possibly related to oligomerization of OA. We did not find many tar balls in the CLARIFY and ORACLES campaigns, with the exception of filters corresponding to RF10 and RF11, which were aged for 1 and 2 days, respectively. RF10 had very viscous aerosol but was mixed with considerable amounts of nitrogen and sulfur. This suggests a removal process, and while there are many unknowns regarding loss processes for tar balls, precipitation near the coast or heterogeneous, photolytically driven processes which may affect the solubility or volatility of tarballs as they are advected west over the ocean may contribute to their removal. Posfai et al. (2003) also reported a dearth of tarballs when sampling in the haze layers representing aged BB plumes, without a clear explanation for their absence.

### 3.2.2 Potassium salts and black carbon

More than 60% of particles, by number, from the two campaigns were potassium salts, either externally or internally mixed. Only K-salts which appeared solid were counted in this number. If a particle was OA with K present, but without a visible K-salt inclusion, this would not be counted as a K-salt. If a particle was BC with a K-crystal attached, this would be counted as a BC-Ksalt internally mixed particle. This is consistent with findings from (Li et al., 2003) where organic particles and potassium salts were the predominant particle types in the smoke. The salts were often mixed with black carbon, organic, or sulfates. Inorganic salts in BBA can result from volatiles from the burning source depositing inorganics onto particles in the BB plume (Jahn et al., 2020; Li et al., 2003; Gaudichet et al., 1995). Different salts will indicate different processes; K-salts will form due to evaporation of potassium in the fire and subsequent near field condensation onto the BC; while this will occur with some S and N as well, co-emitted $SO_2$ and $NO_2$ can oxidize and condense and lead to additional coating in the far field. One common particle type was potassium salt internally mixed with BC, where the K-salt encapsulates the black carbon in a core-shell configuration. EDX analysis can ablate the salt and leave the refractory black carbon core intact. Another common particle type was organic aerosol with interstitial salts. These two common K-salt mixtures are shown in Figure 4. The coating of BC gives rise to absorption enhancements as discussed by Taylor et al. (2020), where they found universally thickly coated BC, and almost no externally mixed BC. The source of the coating is not described in that paper; the TEM results show that a common coating type is a hygroscopic salt, with implications for both absorption enhancement and enhanced CCN capability of the particles.

The three common black carbon mixing states, BC with salt, BC with organic, and externally mixed BC are shown as a fractional amount that exists in each campaign and BL/FT in Table 2. Internal mixing refers to a particle which has two or more separate components, whereas externally mixed particles contain one component per particle. The

predominant mixing state is BC internally mixed with salt, however, BC mixing with organic is likely underestimated due to volatization of organic in the chamber. Table 2 shows a difference between BL and FT in all columns, with the sign of the differences being different in the two campaigns. It should be noted that of the three ORACLES filters collected in the BL, two have marine backtrajectories, so BB organic may be underrepresented here. For CLARIFY, cloud processing may remove the more hygroscopic BC containing particles as these are activated and removed by precipitation, and hence the organic/BC ratio is high relative to the FT, but this does not work for ORACLES. The main finding here is that BC with inorganic, as analyzed by TEM, is the most prevalent BC mixing state.

### 3.2.3 Marine aerosol

CLARIFY and ORACLES aerosol were both influenced by the marine atmosphere. Most CLARIFY filters have sea salt aerosols (SSAs) with Na and/or Cl present in varying ratios in the particles, as presented in Table 3. There are also minor amounts of Ca, Mg, K, as would be present in seawater. Table 4 lists the ORACLES particle percents for Na and Cl as well as ratios. Both Table 3 and Table 4 list altitude, CO, and time from fire source to provide context as to whether the air mass is BB influenced. As a measure of aging and sea salt conversion, Tables 3 and 4 list Na:Cl and Na:S weight percent ratios for particles which have those elements present, averaged per filter. A comparison of Table 3 and Table 4 shows that for ORACLES filters, average Na and Cl wt% is less, per particle, and there is a larger variation in % particles per filter containing Na and/or Cl than in CLARIFY. In ORACLES, Na/Cl per filter is higher than CLARIFY due to the low Cl wt% particle average, and Na/S ratio is generally lower due to the lower Na wt%.

A schematic of the SSA lifecycle, with representative particles from several CLARIFY filters and example mechanisms for Cl depletion, is provided in Figure 5. Briefly, freshly emitted sea salt, generated from ocean bubbles bursting, has Na and Cl present in a 0.86:1 atomic ratio. However, Na and Cl in the sea salt aerosol rarely are in a 0.86:1 ratio as would be expected from freshly emitted SSA, indicating that the particles have been processed. Natural variability can be present, with Krueger et al.(2003) finding that Cl/Na atomic ratio in sea salts increases with particle diameter. The aging timescale of sea salt also varies depending on the production of $NO_2$ and $SO_2$ and its conversion rate to $H_2SO_4$ and $HNO_3$ since these acids displace the Cl, and these rates will vary by location. The aerosols are processed in the atmosphere, with nitrates and sulfates replacing Cl. S is removed from the atmosphere through oxidation of $SO_2$ in water associated with sea salt particles (Sievering et al., 1991; Miller et al., 1987) as well as cloud processing (Beilke and Gravenhorst, 1978), and N species like $HNO_3$ and $NO_2$ are also available for reactions with sea salt. Variations of Na:Cl, then, can help to determine relative aging for SSA, as has been used for example in Kirpes et al. (2018), Hand et al. (2010) and Young et al. (2016). For context, Na and $SO_4$ weight percent in sea salt, based on the composition of sea water, are 60.31 and 7.68, respectively (Seinfeld and Pandis, 2012). Using the atomic weights of sulfur and oxygen, this leaves an expected Na:S ratio of approximately 16:1 in sea salt. A lower than 16:1 ratio indicates Cl displacement by S and is and indicator of aerosol aging, therefore based on the ratios in Table 3, our

samples are aged sea salt. Prior work has shown variation of up to 13% in the atomic percent of S in fresh SSA (Ault et al., 2013).

For CLARIFY, all SSA on filters collected in the MBL have NaCl with varying levels of Cl depletion. The presence of Na colocated with Cl on all below-cloud filters and in only 2/7 of the above-cloud filters suggests the particles are

less aged in the BL samples compared to the FT samples. Gold 23, a filter sampled in the FT, has a high Na:Cl ratio of 20.2 for particles with both Na and Cl, and this suggests that these salts are aged due to the Cl depletion. The other filter sampled above-cloud with Cl, Gold8, has mostly Cl-only particles and a also crystals of Na:Cl which appear freshly emitted with a cubic NaCl structure. Gold 8 has particles similar in morphology to Cl-rich particles present on filters Gold 14, 15, and 18 which will be described in a later section, the difference being that Gold 14,15 and 18

filters did not have any Na-containing particles present. Cl-only particles in the FT suggests mixing of the MBL and FT, as it shows that Cl has reached the FT through turbulent mixing at the top of the MBL. Both Gold8 and Gold23 filters have backtrajectories which show airmasses from the continent which are entirely within the FT. As deep convection does not occur in this region, marine salts which are observed more than a few hundred meters above the BL height in the Ascension Island region of approximately 2250m (Haywood et al., 2021) may be brought in from

outside the region.

In the above-cloud CLARIFY samples, all particles were subject to BB influenced air based on CO values, where we choose 120 ppbv as a CO level indicating BB influenced air above background levels. In the FT, Cl was mostly not present or depleted. BC often mixed with sodium sulfates, Cl, and nitrate, and K-salts were often mixed with $NaSO_4$.


Filter Gold 24, interestingly, shows sodium nitrate mixed with black carbon. N can be difficult to detect in EDX spectra as it is between the C and O peaks and can be difficult to deconvolute; therefore the presence of N in the EDX spectra indicates that there is a substantial amount in the particles. Gold 24 was collected above-cloud and in highly BB-influenced air (331 ppbv of CO; 1.9 µg/cm$^{-3}$ $NO_3$). Gold 9, also collected above-cloud in BB influenced air (329

ppbv of CO; 3.1 µg/cm$^{-3}$ $NO_3$), also has sodium nitrate but to a lesser extent than Gold 24. As these are the two above-cloud CLARIFY filters with the highest CO levels, and are also the two filters which show some presence of sodium nitrate, this suggests that BB air may influence the sea salt conversion from NaCl to $NaNO_3$. This is likely due to the emissions of $NO_X$ in  BB plumes (Jin et al., 2021) to form $HNO_3$ and drive Cl out of the sea salt aerosol. There is also an influence of marine air on BC. For example, the bottom center image in Figure 6e shows a black carbon particle

mixed with sodium nitrate from the Gold 24 filter, collected in the FT. The presence of the Na in the FT suggests BB entrainment into the MBL and subsequent mixing of marine air into the FT, or as an as alternate explanation, sea spray mixing into the FT with BC.

There were different morphologies and compositions of the marine salts due to different salt conversion processes:

some large rounded NaCl over 1.5 micron in diameter and Ca, Mg, and Na sulfates and chlorides. Figure 6 shows a few examples of the many different morphologies of marine salts found on CLARIFY filters. Figure 6F is from Gold

1 which was collected in the MBL, and the particle is BC encapsulated in sodium sulfate; this may be due to aqueous processing where sodium sulfate forms around a BC core. This suggests that while in the FT there is BC and K-salts mixed with sodium, BC, if entrained in the MBL, can also be significantly affected by sea salts and mixed with sodium nitrates and sulfates.

Na and Cl are collocated on individual particles in 5/13 above-cloud filters and 2/3 below-cloud ORACLES filters. All below cloud filters had BC present. RF2_1 BC is not mixed with Na or marine salts. RF5_2 and RF7_2 filters have BC mixed with salts containing Na,S and K. We did not observe clear crystalline sea-salt morphologies in either above-cloud or below-cloud samples as observed in CLARIFY filters, and also did not observe Cl-dominant particles.

Three filters from CLARIFY, Gold 14, 15 and 18 were dominated by Cl particles which do not have Na present. Table 5 shows the three Cl dominant filters along with altitude at filter exposure, CO levels, and time from fire, and SI Figure S2 provides altitude and particle counts during the filter exposure time. The Cl-rich filters are interesting because of the high spatial density as well as the uniformity of the particles on the filter. The uniform particle composition and morphology of the particles suggests that, per filter, the particles have been subject to similar atmospheric processing. Gold 14 was collected in the FT and backtrajectories show interception with biomass burning six days prior to filter collection. Backtrajectories for filters Gold 15 and 18, both collected in the MBL, show that they do not intercept with the fires and are not as influenced by BB air as Gold 14. The presence of the elements Si, K, Ca and Mg are hypothesized to be from SSA rather than biomass burning based on the unique morphology and composition of the particles on these filters. The atmospheric implications for the Cl-rich particles are not clear, although their ability to uptake water may be conducive to their ability to act as CCN.

Gold 14 was collected above-cloud, with 47/47 of the particles having strong Cl and N peaks, and some of the particles show very minor amounts of Si. Wu et al.'s (2020) analysis of the CLARIFY campaign noted an increase in nitrate mass concentration with increasing altitude, and found that the nitrate aerosol mostly existed as ammonium nitrate in the FT. They suggest that the increased levels of nitrate in the FT may be due to colder temperatures at high altitudes which would help partition the $HNO_3$-$NH_3$-$NH_4NO_3$ system into the aerosol phase (Wu et al. 2020). Particle counts from the condensation particle counter (CPC) show a particle count average of 1437 particles/$cm^{-3}$. As ammonium nitrate is a hygroscopic inorganic salt which can dissolve into the aqueous aerosol phase, we hypothesize that the Cl aerosols are from HCl in the gas phase which partitions into the aerosol water and ammonium nitrate aerosol. Relative humidity at filter collection times according to backtrajectories is in the 60-70% range, which would facilitate aerosol deliquescence and the subsequent uptake of HCl. Previous work (Semeniuk et al., 2007; Jahn et al., 2021) has also shown BBA to uptake water in this RH range. Hydrochloric acid partitioning into aerosol water has also been inferred in urban atmospheres, where the highly water absorbing and soluble chloride in the aqueous phase enhances aerosol water uptake through co-condensation and particle growth, causing haze to form (Gunthe et al., 2021). The presence of ammonium in the FT and Cl in the samples, as well as strong Cl and N peaks in the EDX spectra suggest that these particles may be $NH_4Cl$.

Gold 15 was collected below cloud at 329 meters and 21/21 of the analyzed particles were composed predominantly of carbon and chlorine with small amounts of silicon and potassium; the particles appeared similar across the filter so EDX on additional particles was not performed. These particles may show HCl gas uptake onto sea spray aerosol. Gold 18 was collected below cloud at 332 meters. 32/32 particles sampled on the TEM filter were either $CaCl_2$ or $MgCl_2$ with silicon and may have formed through chlorine reacting with Ca and Mg in aerosolized sea water.  Prather

et al. (2013) observed that long chain bioorganic species as well as Ca and Mg form stable collapsed structures as sol-gels, and potentially particles from Gold 15 and Gold 18 may be sol-gel  with Ca and Mg dispersed within a sol gel structure.  The lack of an observable counterion in the EDX spectra of Gold 15 particles may be due to a weaker N signal than in Gold 14, or potentially Cl dispersed within a sol-gel network.

The high density of the chloride particles on the filters, compared to the other TEM filters, along with the uniform composition and morphology of these particles, suggests either condensation onto new particles formed in the FT or condensation onto marine particles formed from spray in the MBL.  While the prevailing view is that new particle formation rarely occurs over open oceans, work by Zheng et al. (2021) shows new particle formation in the remote MBL occurs frequently after the passage of a cold front, with factors such as removal of existing particles by

precipitation, vertical transport of reactive gases from the ocean and cold temperatures facilitating new particle formation.  Iodine species can also form new particles in pristine regions (He et al., 2021).  These conditions do not characterize our sample, however, these studies are part of a growing area of work which supports new particles formation in the remote MBL; if the newly formed particles are aqueous, they can support HCl uptake and may explain the Cl-dominant aerosol observed.  The particles could also originate from sea spray without additional processing,

and as primary particles may be sol gel structures of bioorganics and as observed in Prather et al. (2013).

       The filters were collected in both above and below cloud conditions, which suggests a mixing of the MBL and FT, as Cl, either in the gas or aerosol phase, reached the FT after having being emitted from the ocean; air exchange at the top of the BL due to turbulent mixing may be for the cause of this, particularly as there is a transition from

stratocumulus to cumulus near Ascension Island, and a corresponding increase in convection (Gordon et al., 2018). We hypothesize that Cl in the FT may be from HCl in the BL which gets taken up into cloud droplets, and on their evaporation at cloud top is re-released onto more neutral aerosol such as ammonium nitrate.

**3.3 Marine boundary layer and aqueous phase processing**

       The MBL had a large effect on BB aerosols and as well as processing of sea salt aerosols.  A comparison of cloud processing and time in the FT and MBL for the aerosol collected during the filter sampling time is provided in Figure 7.  The filters are segregated by collection in the MBL and FT and the fractional time spent in each environment is

shown in Panels B (collected in BL) and D (collected in FT).  Panel D shows that aerosols collected in the FT,

irrespective of campaign, have spent nearly all their time in the FT. Panel B shows that of filters sampled in the BL, CLARIFY aerosols have spent more than 80% of the fractional time in the BL in the day before filter sampling compared to ORACLES' 45%.

The cloud processing intensity, which is the mean cloud liquid water content multiplied by the mean in-cloud time, in the days prior to the airmass reaching the downwind filter collection location is provided, in Figure 8 Panel A. Cloud processing intensity is a metric described in detail in Che et al. (submitted). Aerosols collected in the MBL were subject to more cloud processing than those collected in the FT, as the aerosols would need to be entrained into BL. ORACLES aerosol, on average, spent slightly more time in-cloud and with clouds with a higher liquid water content

than CLARIFY aerosol, particularly apparent for those filters collected in the MBL.

In sea salt, K weight percent is 1.1, assuming sea salt has the composition of seawater and ignoring atmospheric processing (Seinfeld and Pandis, 2012). The K-salt from BB has a significantly higher weight percent than the minor amounts of K found in seawater and sea salt. The fraction of K-salt mixed with sodium increases with age, as shown

in Figure 7 Panel C. This implies that the K-salt from biomass burning is processed in a way which allows Na incorporation into the particle. As Na is not volatile at atmospheric pressures, we hypothesize that the mechanism is processing by particle mixing, either by cloud drop coalescence (Grabowski and Wang, 2013) or drizzle wash out of aerosol and evaporation.

The interaction of the MBL with BBA and the effect of BB air on marine aerosol has been shown in previous sections, and we hypothesize the mixing of Na salts with BBA to be due to aqueous processing and particle mixing through, for example, cloud drop coalescence. An aqueous K-salt particle from CLARIFY's Gold 9 filter is presented in Figure 8. It has varying Na:K ratios by weight percent, designated in the red circles, and there is no Cl in the particle. We assume that the Na present is from marine sources because the area with an 11:1 ratio has 13 wt% Na but only 1.2

wt% K; this is a higher Na wt% than would typically be expected from BBA. However, it should be noted that Na may also be from biomass burning, as sodium has been noted in BB fuel, with the type influencing the amount of Na in particles (Hudson et al., 2004). The image shows that Na mixes with K-salts, organic and sulfur in varying degrees throughout a single aqueous particle and therefore this is a potential mechanism for Na incorporation into existing K-salts.

Single particle structure and morphology is important as it affects aerosol optical properties and ability to act as cloud condensation nuclei. Figure 9 shows BC + K-salt (top panel) and BC + (organic/sulfur) mixtures as a function of time in cloud encountered one day before sampling. The spherical nodules are black carbon and the more reflective white areas in the bottom panel are the sulfur/organic mixture. Each particle ID shows the TEM image and associated time

in cloud (hours) in the day before sampling, the mean liquid water content of the clouds (g/kg) in the day before sampling, and the weight percent of all elements other than carbon in the particle. From left to right, the time in-cloud increases. If the electron beam visibly altered the particle, an "after" image is shown along with an "before" image to

indicate the particle's response to the electron beam. In each particle, the black carbon is insoluble and appears unaffected by increased cloud processing; however, both the K-salts and the sulfur/organic appear affected by

increased time in cloud. In the left most panel for Figure 10, top panel, the K-salt is completely unaffected by exposure to the electron beam. The other images in Figure 10 show electron beam damage, indicating that these K-salt structures are more susceptible to volatilization and degradation due to the electron beam; this may be due to degradation of the K-salt during cloud processing and the amorphization from hydration, dissolution, and recrystallization of the salt during processing  As an alternative explanation, nitrates and sulfates have been shown to be affected by electron

beam exposure after exposure to acid gases or water vapor (Jahn et al., 2021; Hoffman et al., 2004). The K-salt particles on the right, which were subject to the most time in cloud, have a less distinct morphology which may indicate periods of water uptake and loss. In Figure 9, bottom panel, the sulfur-organic constituent of the aerosol which was subject to the least amount of time in-cloud appears to have high viscosity, and was unaffected by the electron beam. After more time in cloud, the sulfur/organic on these particles appear to have a lower viscosity, to be more flat, and

also to be more affected by the electron beam. Although these are only a few particles, the images imply that cloud processing can affect certain constituents of mixed aerosols more than others; the structure of the K-salt and the viscosity of the sulfur/organic mixture appear to be more affected by aqueous processing compared to BC.

**3.4 Elemental mixing in individual particles**


EDX of individual aerosol particles can provide information on elemental mixing across the entire ORACLES and CLARIFY sample sets. Figure 10 shows the elemental mixing of CLARIFY and ORACLES samples collected in the free troposphere and boundary layer, designated in separate columns, with each row of pie charts indicating whether a set of two elements are colocated on individual particles for all particles of the specified particle set. All percentages

are based on particle numbers rather than mass. The elements S, Na, Cl and K were chosen for elemental mixing analysis as they are the elements which most commonly appear in the EDX spectra. C and O are found in the spectra of almost all particles, and so these elements are not included in this analysis. Supplementary Information, Figures S3 and S4, also includes BL and FT Na-S-K and Na-S-Cl ternary diagrams for additional context, showing that there is more Na and Cl in the CLARIFY samples, particularly the BL.


BBA is more diluted on CLARIFY than ORACLES filters, and S/K particle fractions in the BL/FT suggest increased mixing due to detrainment and entrainment for CLARIFY samples. A higher fraction of ORACLES aerosol contains S, 80%, compared to CLARIFY's 40%. S is mixed with K in 34% of CLARIFY and 73% of ORACLES aerosols. This suggests that ORACLES S is predominantly from biomass burning as it is colocated with K, and potassium is

frequently used as a marker for BBA, consistent with the high amounts of K-salts observed in TEM samples. 76% of ORACLES FT aerosol contain colocated  S + K, compared with 59% of ORACLES BL aerosol. In comparison, 35% and 33% of CLARIFY FT and BL aerosols have colocated S+ K, respectively, implying a dilution of BBA in the CLARIFY aerosol population collected on filters in comparison to ORACLES. The similar fraction of S/K particles in the FT and BL during CLARIFY as compared to ORACLES may be due to the transition over the ocean and the

increased entrainment and detrainment across the BL top.  Higher fractions of S-containing particles in ORACLES may also be related to cloud processing (Ervens et al., 2018), with aqueous formation pathways for sulfate in cloud water predicted to be faster than gas-phase formation pathways.

More CLARIFY than ORACLES particles show evidence of marine influence as evidenced by the presence of Na

and Cl.  70% of CLARIFY aerosols and 33% of ORACLES aerosol contain some combination of Cl and/or Na.   Of ORACLES aerosols containing Na, only 2% are mixed with Cl, compared with 20% of CLARIFY particles. Cl can be found in fluids of some vegetation and so can be present in BBA (Liu et al. 2000), thus while ORACLES Cl may be due to either biomass burning or a marine influence, it is likely that the high fraction of CLARIFY particles with Cl present, relative to the ORACLES population, is indicative of a marine influence for particles in the BL and lower

altitudes in the FT.  This is also supported by the difference in CLARIFY Cl in the BL (75% of particles) versus the FT (31% of particles), as well as CLARIFY filters Gold 14, 15 and 18 which were dominated entirely by Cl particles. Cl particle mixing differences between BL/FT dominate in CLARIFY aerosol, but not in ORACLES, as evidenced by K+Cl and S+Cl mixing.  Cl is present in 6% and 47% of ORACLES and CLARIFY particles, respectively.  K+Cl is present in 3% and 8% of ORACLES FT and BL particles, and 10% and 42% of CLARIFY FT and BL particles,

respectively.  S+Cl is present in 3% and 4% ORACLES FT and BL particles, and 8% and 27% of CLARIFY FT and BL particles, respectively.  CLARIFY Cl is mixed with K or S three to four times more in the BL than in the FT, while the BL does not have as much of an influence on Cl mixing with K or S for ORACLES particles.   The increased Cl mixing with S/K in the CLARIFY BL may be due to secondary processes which deposit Cl onto BBA, or a high fraction of primary SSA which also includes S and/or K.


The reverse trend is observed for Na, with Na particle mixing differences between BL/FT more apparent in ORACLES than CLARIFY, as evidenced by K+Na and S+Na mixing.  44% of CLARIFY aerosol contains Na compared to 30% of ORACLES aerosol. K+Na is present in 22% and 50% of ORACLES FT and BL particles and 35% and 43% of CLARIFY FT and BL particles, respectively.  S+Na is present in 24% and 46% of ORACLES FT and BL particles

and 28% and 32% of CLARIFY FT and BL particles, respectively.  Thus while Na is slightly more likely to be mixed with K or S in the CLARIFY BL compared to FT, Na is mixed with S or K about two times more in the ORACLES BL than FT.  This may be due to increased BL/FT entrainment and detrainment for CLARIFY samples, and more in-cloud aqueous processing and particle mixing for ORACLES samples which can deposit Na onto BBA in the BL.

**4.0 Conclusions**

As CLARIFY sampled older smoke than ORACLES, this study, by comparing the two campaigns in the FT and BL, shows ways in which African BBA smoke is affected by SSA and marine air, and reciprocally how SSAs may be affected by mixing with BB plumes.  Single particle analysis revealed considerable heterogeneity in the mixing and

processing of CLARIFY and ORACLES aerosols.  The main aerosols are BBA and SSA.  We found similarities to the previous major campaign which analyzed African BBA, SAFARI-2000, in that we observed an abundance of

potassium salts, black carbon, and organic carbon with interstitial potassium salts (Pósfai et al., 2003; Li et al., 2003). Posfai et al. (2003) as well as our analyses show considerable internal mixing between black carbon and salts; therefore, this suggests that the thick coating on CLARIFY and ORACLES BC as measured by the SP2 (Wu et al. 2020; Redemann et al 2021) is often due to inorganics rather than organics. This has implications for radiative effects due to lensing, as well as CCN capability as the inorganic salts internally mixed with BC would increase the particle's hygroscopicity. This is important as it suggests that the salts formed in the fire via evaporation and recondensation drive the mixing of the carbon aerosol as the secondary inorganic condenses, and that the organic fraction is separate. This is consistent with findings regarding emissions of BC and K-salts and other salts in the flaming phase of a fire, while organic emissions occur during the pyrolysis or smoldering phases (Haslett et al., 2018). Field conditions including different burning material and wet and dry conditions can also lead to spatial and temporal variability which may affect he near-source mixing of fresh BBA, These findings are caveated due to the loss of organic in the TEM chamber, which would artificially increase the internally mixed and salt-BC fraction, particularly for CLARIFY samples.

While SAFARI analysis found a large number of KCl salts near source, both the CLARIFY and ORACLES campaigns sampled much more aged aerosol, and did not find many KCl particles. This is presumably because replacement of Cl by nitrates and sulfates has occurred prior to ORACLES and CLARIFY BBA sampling. While the SAFARI campaign and other recent biomass burning campaigns found tar balls, our TEM analysis did not find tar balls other than on filters RF10 and RF11, which were aged for approximately 1 and 2 days, respectively. This finding implies a reduction in tar balls in aged African BB plumes. Tar ball incorporation in BB models (Jacobson, 2014) have been hindered due to lack of data, as tar balls can only be definitively detected with time-intensive single particle electron microscopy. While other work shows that tar balls are a significant fraction of BB aerosol, with some showing that tar balls outnumber BC by a factor of 10 (Hand et al., 2005; China et al., 2013) our analysis shows a lack of tar balls in the aged BB plumes, consistent with Posfai et al. (2003) who also reported a dearth of tar balls in aged plumes as a puzzling phenomenon. Since tar balls are a light-absorbing particle, the absorption from aged plumes is dominated by non- tar ball components like BC, brown carbon, and dust; the absence of tar balls in these aged plumes can help constrain models on radiative forcing in the region.

There were differences observed between FT and BL aerosol. More BBA was found in the FT, and more SSA was found in the BL, as would be expected; as BBA aerosol aged, there was increased entrainment into the BL for the CLARIFY samples relative to the ORACLES samples. SSA in the MBL was less aged than those sampled in the FT, as measured by Cl depletion. The presence of Cl and, on some filters, freshly emitted NaCl in the free troposphere suggests exchange of the MBL and FT in some regions through turbulent mixing at the top of the BL. For ORACLES, S-containing and K-containing particles were much more likely to be mixed with Na in the BL as compared to the FT. These BL/FT differences with Na-S and Na-K particles were not as evident in CLARIFY. In CLARIFY, the largest composition changes between the FT and BL were that S-containing particles were more likely to be mixed with Cl in the BL as compared to the FT, and similarly, K-containing particles were more likely to be mixed with Cl in the in

the BL compared to the FT. These Cl trends were not as clear in ORACLES, and only a minor fraction of ORACLES particles contain Cl. This may suggest that for ORACLES, aqueous processing is a key driver in depositing Na onto BB particles, for example by droplet coalescence in clouds and drizzle evaporating before it reaches the surface. For CLARIFY, secondary processes may be important in depositing Cl onto BB particles.

We found evidence of BBA interaction with the MBL as well as marine salts affected by BB air, which has not been reported to date in these campaigns. Mixing of marine air with BB air affects sea salts because of Cl replacement by nitrate, as BB plumes have elevated levels of NOx which replaces the Cl in SSA. Reciprocally, BBA was influenced through, for example BC mixing with sodium sulfates and sodium nitrates and the presence of Na and Cl on large number of BB particles. There is considerable mixing of Na with BBA through what we believe is aqueous processing of particles, through, for example either cloud drop coalescence or drizzle washout of aerosol and subsequent evaporation. Na and Cl were present in a higher fraction of particles in CLARIFY samples as compared to ORACLES. TEM particles show evidence of aqueous processing, with varying ratios of soluble components throughout a single particle. Particle morphology also show evidence of cloud processing, with soluble components more significantly affected by time in cloud than insoluble components such as black carbon.

The uniformity and ubiquity of Cl-containing aerosol particles on three CLARIFY filters suggests that particles on these filters have been uniformly processed and the existence of areas dominated by Cl aerosol. Further, the high density of Cl spatially on these filters implies HCl condensation onto existing particles which have been recently formed and processed uniformly. Another explanation may be that these are primary particles such as solgel organic particles with Ca, Mg and Cl dispersed. The presence of Cl rich particles are interesting as they may act as good CCN due to their ability to uptake water.

TEM analysis indicates either a loss of organic, including as an organic coating, with plume age, or an increase in OA volatility with BB plume age. The reduced amount of organic observed in CLARIFY compared to ORACLES may be due to less organic being present, a more volatile organic which evaporated in the TEM chamber, or both. Che et al. (submitted) noted SOA formation in the first ~70 hours of BB aging for ORACLES; our results suggest that the secondary organic which forms may be more volatile than the initial organic emitted from biomass burning. There have been different explanations for the vertical structure of single scattering albedo near Ascension Island, with Taylor et al. (2020) and Wu et al. (2020) suggesting a partitioning of inorganic ammonium nitrate onto existing particles at colder temperatures, and Dobracki et al., (2022) hypothesizing that the single scattering albedo differences are due to scattering organic material that is lost from BBA. Sedlacek III et al. (in prep) also found a loss of organic material coating BC with plume age in ORACLES; further the authors define different regimes for BBA where organic coating on black carbon increases in the first few hours after emission, the coating mass then plateaus after a several hours to several days when there are competing chemical physical processes such as photochemistry, SOA production, fragmentation and oxidation, and after several days of aging there is material loss due to cloud processing, volatility, and bleaching of brown carbon. While there are different explanations for the vertical structure of single scattering

albedo near Ascension Island as well as mechanisms for organic loss with age, our results indicate that higher volatility organic is associated with aging; therefore we hypothesize that fragmentation of carbon chains, either through photolysis or oxidation of the organic, is the predominant mechanism for the apparent higher volatility of aged organic. Our results are compatible with Dobracki et al.'s (2022) and Sedlacek III al.'s (submitted) findings of a loss of organic

with age. Increasing oxidation of BB with age, as noted by Wu et al. (2020), and more volatile organic or loss of organic as observed with TEM results implies fragmentation of carbon chains.

Due to the changes in organic aerosol and the noted effects of the MBL on BBA, it appears that aqueous processing, oxidation, photolysis, evaporation, condensation, and interaction with the MBL are key drivers in physical and

chemical properties such as mixing state and elemental composition of aged BBA. Studies on combustion from different fuel and burn phases show differences in primary particle types in terms of mixing, amount of organic and elemental composition, for example for particles formed in the flaming and smoldering phase (Liu et al. 2017). Our results imply that due to the cloud and aqueous processing that African BBA is subject to, the salt phases present in the BBA will affect the ability of the particles to uptake water, act as CCN, and undergo aqueous and cloud processing.

Therefore the inorganic salt content of fresh BBA, fuel type and burning conditions, as well as gas phase oxidation of NOx leading to formation of $NO_3$ as a significant pathway for further addition of inorganic salts, are key components for the atmospheric aging of BBA in these regions.

**Data Availability**

Data are publicly available at the ORACLES and CLARIFY archives: http://data.ceda.ac.uk/badc/faam/data/2017 and https://espo.nasa.gov/oracles/archive/browse/oracles/id14. Data not on these websites can be provided by request.

**Author contribution**

MS, H. Coe, and CD designed the research. JH, H. Coe, JR, PZ, AN are PIs of the campaigns. JT, MS, PW, SP, PZ, AN, SH performed field work or provided support for filter collection. SH and MS designed the sampling apparatus. CD performed laboratory analysis. CD and H. Che provided figures. CD, H. Che, MS and LZ analyzed data sets. CD led the paper writing, and all co-authors contributed to ideas and writing.


**Competing interests**

JH, PZ, and PF are guest editors for the ACP Special Issue "New observations and related modelling studies of the aerosol–cloud–climate system in the Southeast Atlantic and southern Africa regions". The remaining authors declare that they have no conflicts of interests.


**Acknowledgements**

The first author was supported by the NASA Postdoctoral Fellowship Grant. ORACLES is a NASA EARTH Venture Suborbital-2 investigation, funded by the US National Aeronautics and Space Administrations (NASA)'s Earth Science Division and managed through the Earth System Science Pathfinder Program Office (grant no.

NNH13ZDA001N-EVS2). CLARIFY-2017 was a Natural Environment Research Council (NERC) Large Grant NE/L013584/1. HC and MS are supported by a Department of Energy (DOE) Atmospheric System Research (ASR) grant DE-SC0020084. LZ is funded by a Tel Aviv University postdoc fellowship. PF is supported by the AErosols, RadiatiOn and CLOuds in southern Africa (AEROCLO-sA) project funded by the French National Research Agency under grant agreement n° ANR-15-CE01-0014-01, the French national programs LEFE/INSU and PNTS, the French

National Agency for Space Studies (CNES), the European Union's 7th Framework Programme (FP7/2014-2018) under EUFAR2 contract n°312609, and the South African National Research Foundation (NRF) under grant UID 105958. PZ acknowledges additional support from Department of Energy grant DE-SC0021250. We thank George Levi, instrument scientist at Tel Aviv University, for his expertise in TEM operation and analysis. Further we thank the manuscript reviewers for their time and comments.

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

**Table 1 Filter IDs, ancillary online aerosol data, location, altitude, backtrajectory-based time from fire as detailed in the text, and in-cloud time over previous 24 hours**

| Campaign and year | Filter | Date | Particles analyzed | Latitude (°) | Longitude (°) | Altitude (m) | Org (µg /cm$^{-3}$) | SO$_4$ (µg /cm$^{-3}$) | NO$_3$ (µg /cm$^{-3}$) | NH$_4$ (µg /cm$^{-3}$) | BC (µg /cm$^{-3}$) | BC (particles /cm$^{-3}$) | CO (ppbv) | Cloud time in 24hrs (hours) | Above or Below cloud | Time from fire (days) |
|---|---|---|---|---|---|---|---|---|---|---|---|---|---|---|---|---|
| ORACLES 2017 | RF11Filter5 | 8/30/2017 | 47 | -9.47 | 5 | 3505 | 20.5 | 1.5 | 3.1 | 1.3 | 3.8 | 1162 | 395 | 0 | Above | 2 |
| ORACLES 2018 | RF02_1 | 9/30/2018 | 23 | -7.64 | 5 | 894 | 1.1 | 0.5 | 0.1 | 0.1 | 0.2 | 56 | 110 | 9.35 | Below | 5 |
| | RF02_2 | 9/30/2018 | 35 | -7.82 | 5.03 | 2606 | 6.6 | 0.9 | 0.3 | 0.2 | 0.9 | 273 | 210 | 6.55 | Above | 1 |
| | RF03 | 10/2/2018 | 59 | -7.67 | 5.5 | 982 | | | | | 1.2 | 346 | 156 | 18.08 | Above | 6 |
| | RF04 | 10/3/2018 | 65 | -6.75 | 7 | 1195 | 0.5 | 0.4 | 0.0 | 0.1 | 0.3 | 117 | 120 | 6.58 | Above | 6 |
| | RF05_1 | 10/5/2018 | 55 | -9.5 | 6.17 | 943 | 0.7 | 0.5 | 0.1 | 0.2 | 1 | 297 | 154 | 11.29 | Above | 6 |
| | RF05_2 | 10/5/2018 | 64 | -9.5 | 6.21 | 378 | 0.2 | 0.2 | 0.0 | 0.1 | 0.5 | 119 | 106 | 8.42 | Below | marine |
| | RF05_3 | 10/5/2018 | 37 | -9.5 | 6.11 | 3247 | 6.4 | 1.2 | 0.5 | 0.4 | 0.9 | 294 | 210 | 0 | Above | 1 |
| | RF06_1 | 10/7/2018 | 49 | -8.91 | 5 | 2444 | 6 | 1.1 | 0.4 | 0.4 | 1.3 | 421 | 248 | 0 | Above | 2 |
| | RF06_2 | 10/7/2018 | 39 | -6.86 | 5 | 2570 | 2.3 | 0.6 | 0.1 | 0.2 | 0.5 | 193 | 173 | 0 | Above | 2 |
| | RF07_1 | 10/10/2018 | 43 | -12.77 | 5.01 | 1091 | 0.6 | 0.3 | 0.0 | 0.1 | 0.5 | 159 | 121 | 2.59 | Above | 6 |
| | RF07_2 | 10/10/2018 | 29 | -7.39 | 5 | 159 | 0.3 | 0.2 | 0.0 | 0.0 | 0.4 | 108 | 123 | 0.18 | Below | marine |
| | RF09 | 10/15/2018 | 56 | -11.35 | 5 | 1307 | 1.2 | 0.5 | 0.1 | 0.1 | 1 | 265 | 158 | 0.5 | Above | 7 |
| | RF10 | 10/17/2018 | 66 | -7.18 | 10.5 | 1986 | 18.5 | 3 | 2.6 | 1.4 | 2.3 | 807 | 417 | 2.25 | Above | 1 |
| | RF11 | 10/19/2018 | 62 | -7.95 | 9 | 3027 | 3.7 | 0.6 | 0.3 | 0.2 | 0.9 | 292 | 190 | 0.08 | Above | 2 |
| | RF13 | 10/23/2018 | 33 | -5.01 | -0.68 | 1127 | 0.1 | 0.1 | 0.0 | 0.0 | 0.1 | 42 | 118 | 6.94 | Above | 4 |
| CLARIFY 2017 | Gold_1 | 8/17/2017 | 49 | -8.8 | -11.52 | 323 | 4.1 | 1.9 | 0.2 | 0.7 | 0.5 | 195 | 108 | 0 | Below | marine |
| | Gold_8 | 8/22/2017 | 27 | -8.46 | -13.43 | 3902 | 6.9 | 1.3 | 1.4 | 1.0 | 1.2 | 380 | 204 | 20.77 | Above | 7 |
| | Gold_9 | 8/23/2017 | 39 | -5.67 | -12.42 | 2813 | 18.8 | 2.9 | 3.1 | 2.0 | 3 | 934 | 329 | 0 | Above | 4 |
| | Gold_10 | 8/24/2017 | 42 | -8.37 | -15.24 | 2918 | 3.9 | 0.6 | 0.3 | 0.3 | 0.8 | 232 | 158 | 0 | Above | 5 |
| | Gold_11 | 8/24/2017 | 54 | -7.7 | -13.85 | 319 | 0.3 | 0.3 | 0.0 | 0.1 | 0.1 | 17 | 70 | 0 | Below | 15 |
| | Gold_14 | 8/28/2017 | 47 | -8.26 | -13.74 | 2845 | | | | | | 683 | 262 | 0 | Above | 6 |
| | Gold_15 | 8/28/2017 | 22 | -8.28 | -13.66 | 329 | | | | | 1 | 287 | 158 | 0.3 | Below | marine |
| | Gold_18 | 8/29/2017 | 32 | -8.69 | -12.47 | 332 | | | | | 0.5 | 174 | 119 | 0 | Below | marine |
| | Gold_19 | 8/30/2017 | 57 | -8 | -17.08 | 1969 | 5.7 | 1.6 | 0.9 | 0.8 | 1.8 | 535 | 212 | 0.38 | Above | 7 |
| | Gold_20 | 8/30/2017 | 30 | -8.03 | -17.3 | 329 | 3.6 | 1.2 | 0.2 | 0.5 | 0.7 | 225 | 130 | 0 | Below | marine |
| | Gold_21 | 9/7/2017 | 43 | -8.32 | -18.48 | 2357 | | | | | 1.5 | 436 | 177 | 0 | Above | 6 |
| | Gold_22 | 9/2/2017 | 43 | -5.66 | -13 | 2139 | 2.1 | 0.6 | 0.2 | 0.3 | 0.7 | 208 | 128 | 0 | Below | 9 |
| | Gold_23 | 9/2/2017 | 44 | -6.14 | -13.52 | 3500 | 12.2 | 1.3 | 2.4 | 1.3 | 2.5 | 750 | 273 | 0 | Above | 4 |
| | Gold_24 | 9/4/2017 | 24 | -7.91 | -12.72 | 1950 | 13.4 | 2.3 | 1.9 | 1.4 | 3.6 | 968 | 331 | 7.84 | Above | 4 |

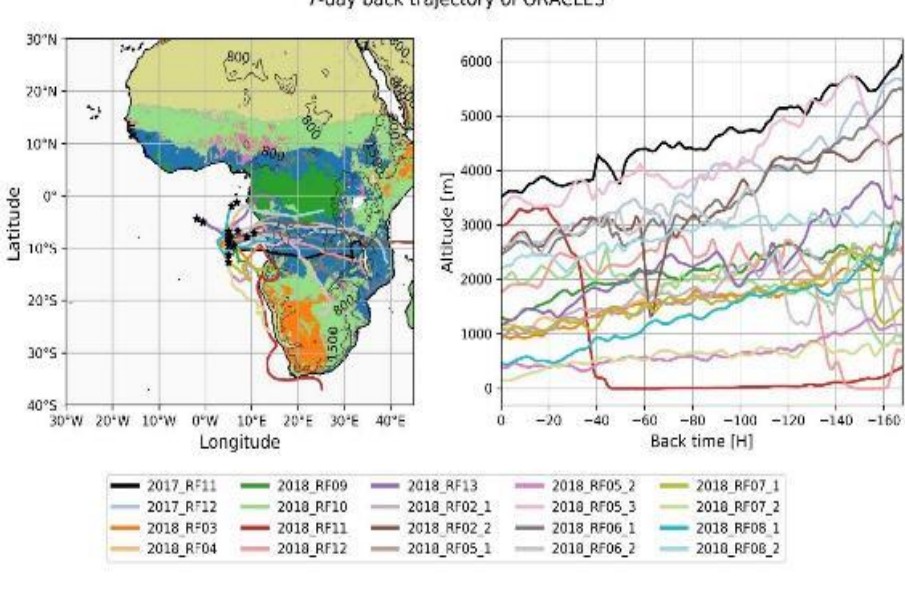


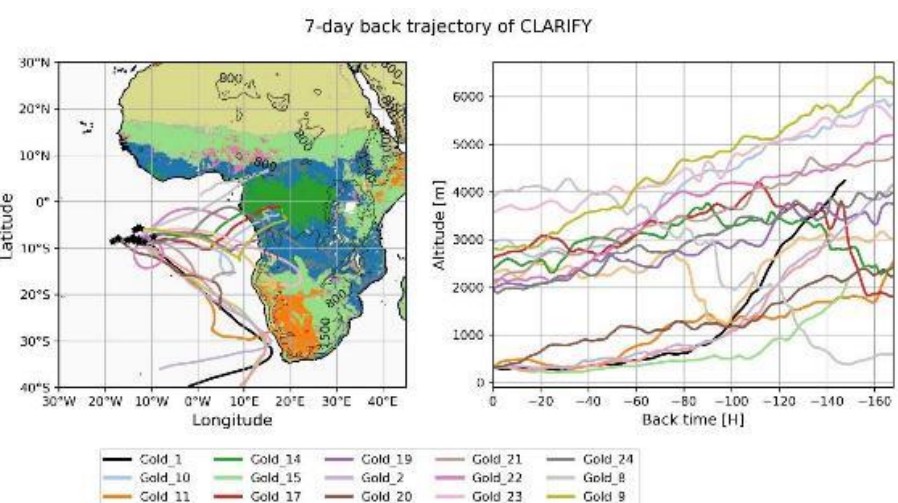

**Figure 1 The location of filter sampling and back trajectories related to each filter, including altitude for ORACLES 2017-18 (upper panel), and CLARIFY 2017 (lower panel). Map colors relate to MODIS land cover types.**


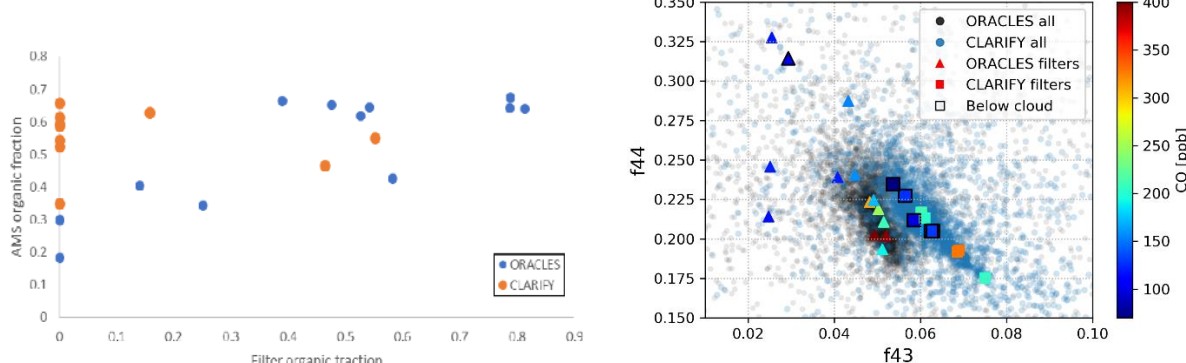

**Figure 2 (left) AMS organic fraction vs filter organic fraction and (right) f44 vs f43 space for ORACLES and CLARIFY campaigns with filters marked as triangles for ORACLES and squares for CLARIFY. Colors of the marks denoting filter sampling represent CO concentration, as shown in the colorbar. Samples collected below-cloud are outlined with a black border.**

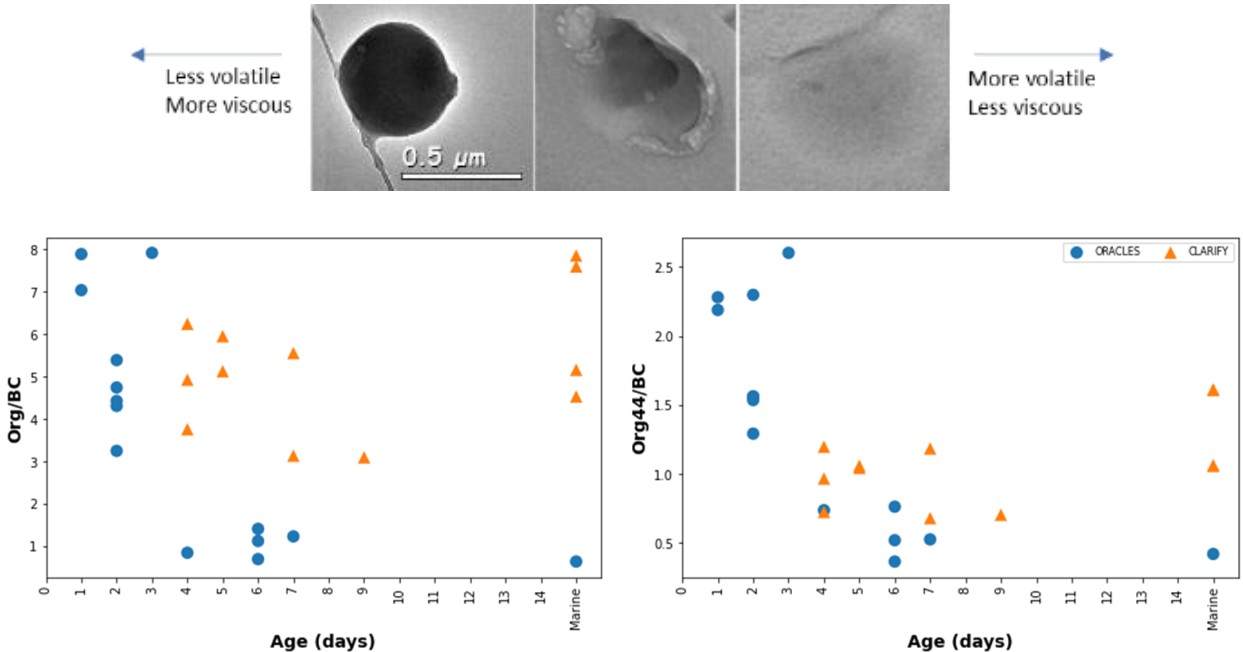

**Figure 3 Example of different viscosity/volatility organic aerosol (top panel) showing more round and viscous particles for ORACLES (on the left), and more volatile for CLARIFY (middle and right top panels) and Org/BC and Org44/BC ratios with time from fire source (bottom panel)**

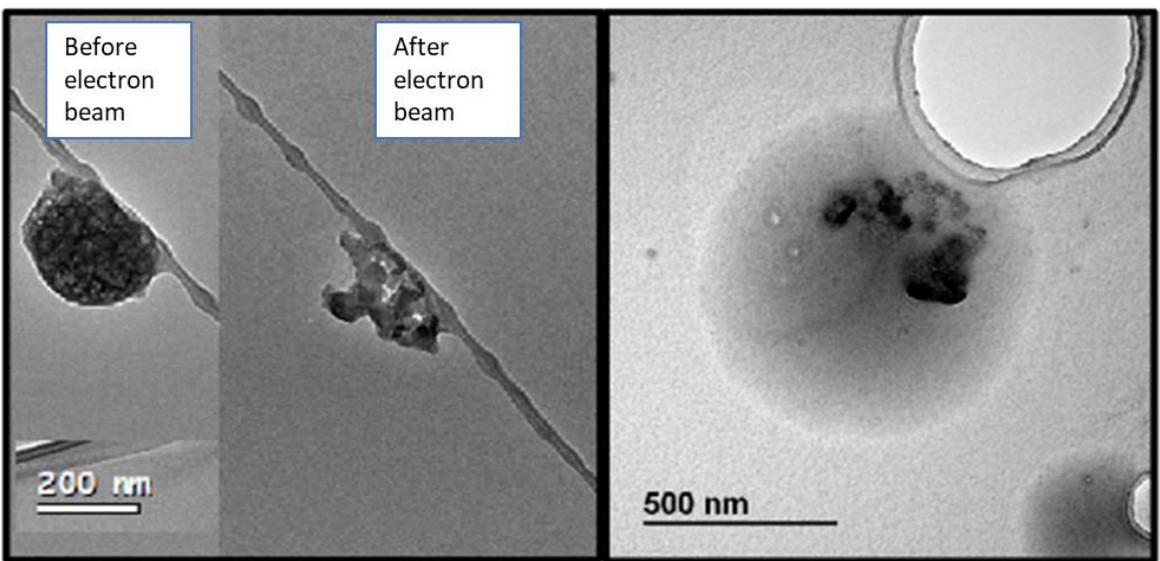


**Figure 4 Potassium salt in a core-shell morphology around a refractory BC core (left) and organic with interstitial K-salt (right). Note the difference in scale between the two images.**

**Table 2 Black Carbon Mixing State by campaign. FT/BL, BC-Salt and BC-Organic refer to internally mixed particles**

|  | BC – Salt | BC – Organic | BC - external |
|---|---|---|---|
| ORACLES BL | 0.78 | 0.00 | 0.22 |
| ORACLES FT | 0.53 | 0.31 | 0.16 |
| CLARIFY BL | 0.50 | 0.29 | 0.21 |
| CLARIFY FT | 0.67 | 0.07 | 0.26 |

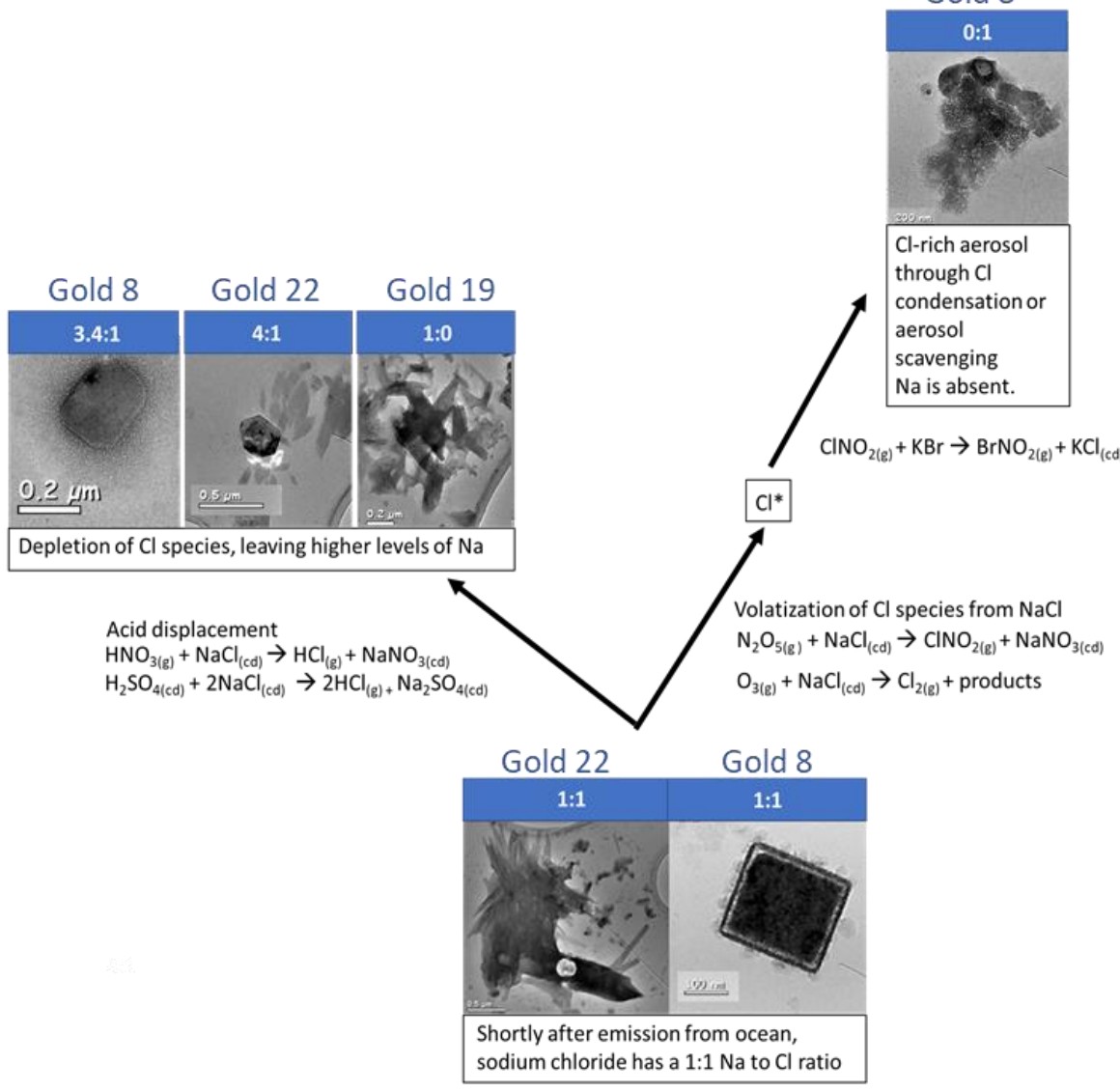

**Figure 5 Schematic showing different stages of sea salt conversion with example mechanisms. The particles are from CLARIFY filters and range from sea salts which have been freshly emitted to Cl-depleted particles containing nitrates and sulfates. Cl aerosol formation is also shown. Na:Cl ratios showing depletion of Cl with sea salt conversion are shown in the bar above each particle image.**

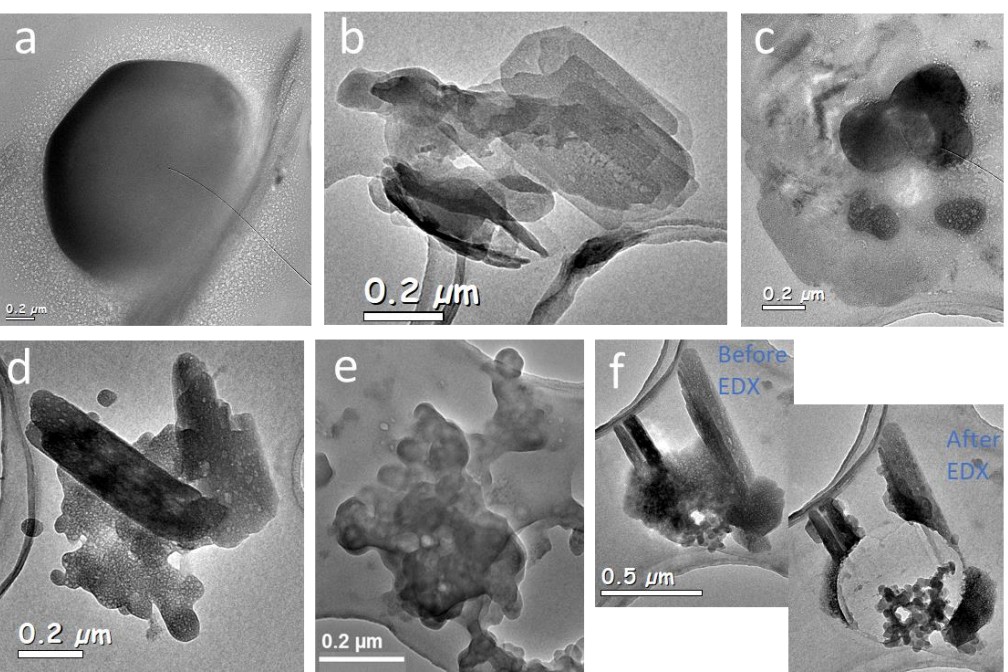

**Figure 6 Assorted sea salt particles. (a) ~1.5 micron rounded NaCl; Gold 1  (b)  CaSO$_4$; Gold 1    (c) NaCl and Mg; Gold 1 (d) Na$_2$SO$_4$ with K; Gold 1 (e) NaNO3 mixed with BC; Gold 24  (f) BC encapsulated in Na$_2$SO4 with K ; Gold1**




**Table 3** Na dominant aerosols on CLARIFY filters with sampling location, CO levels, time from source, and percentage of particles on the filter with either Na or Cl. The ratios represent the weight percent ratio per particle, averaged across all particles with the elements of interest on each filter.

| | Altitude (m) | Above/ Below cloud | CO (ppbv) | Time from fire (days) | Particles with Na or Cl (%) | Avg Na wt% | Avg Cl wt% | Na/Cl | Na/S | Representative particle and filter summary |
|---|---|---|---|---|---|---|---|---|---|---|
| Gold_1 | 323 | Below | 108 | marine | 98 | 10.6 | 10.7 | 7.2 | 5.2 |  Na and Cl with vary levels of Cl depletion; some Na with S; large NaCl particles, BC mixed with NaSO4 |
| Gold_8 | 3902 | Above | 204 | 7 | 100 | 13.7 | 12.6 | 4.3 | NA |  Mostly Cl particles without Na although some NaCl 1:1 crystals |
| Gold_9 | 2813 | Above | 329 | 4 | 37 | 5.8 | | NA | 13.1 |  No Cl present; Na mixed with BC and BB salts, sodium nitrate |
| Gold_10 | 2918 | Above | 158 | 5 | 74 | 5.4 | | NA | 5.4 |  No Cl present; sodium sulfate, BC mixed with Na |
| Gold_11 | 319 | Below | 70 | 15 | 80 | 19.2 | 4.7 | 6.1 | 8.1 |  NaCl, ammonium chloride, sodium sulfate, Na with depleted Cl all present |
| Gold_19 | 1969 | Above | 212 | 7 | 29 | 3.7 | | NA | 2.0 |  Na sulfate, BC mixed with Na sulfate; no Cl, BC with Na, P, S, K, Ca |
| Gold_20 | 329 | Below | 130 | marine | 90 | 3.4 | 0.6 | 12.0 | 3.5 |  BC mixed with Na; NaS crystals; NaCl but most particles Cl depleted |

| | Altitude (m) | Above/Below cloud | CO (ppbv) | Time from fire (days) | Particles with Na or Cl (%) | Avg Na wt% | Avg Cl wt% | Na/Cl | Na/S | | Representative particle and filter summary |
|---|---|---|---|---|---|---|---|---|---|---|---|
| Gold_21 | 2357 | Above | 177 | 6 | 36 | 1.1 | | NA | 0.8 | | Trace amounts of sodium, no marine salts noted |
| Gold_22 | 2139 | Below | 128 | 9 | 77 | 12.5 | 8.7 | 3.9 | 3.3 |  | $Na_2SO_4$, NaCl |
| Gold_23 | 3500 | Above | 273 | 4 | 63 | 7.6 | 0.5 | 20.2 | 3.7 |  | Low levels of Cl, BC mixed with Na, S, K |
| Gold_24 | 1950 | Above | 331 | 4 | 77 | 2.3 | | NA | 1.9 |  | Sodium nitrate mixed with BC |

**Table 4** Na dominant aerosols on ORACLES filters with sampling location, CO levels, time from source, and percentage of particles on the filter with either Na or Cl.  The ratios represent the weight percent ratio per particle, averaged across all particles containing the elements.

| | Altitude (m) | Above/Below cloud | CO (ppbv) | Time from fire (days) | Particles with Na or Cl (%) | Avg Na wt% | Avg Cl wt% | Na/Cl | Na/S | | Representative particle and filter summary |
|---|---|---|---|---|---|---|---|---|---|---|---|
| RF11Filter5 | 3505 | Above | 395 | 2 | 45 | 2.2 | | | 1.1 | | |
| RF02_1 | 894 | Below | 110 | 5 | 22 | 0.4 | 0.7 | | 0.3 |  | BC, K-salt, Ca bearing |
| RF02_2 | 2606 | Above | 210 | 1 | 49 | 1.1 | 0.5 | 2.0 | 0.8 |  | BC, Ca bearing |
| RF03 | 982 | Above | 156 | 6 | 0 | | | | |  | BC mixed with salts and sulfates |
| RF04 | 1195 | Above | 120 | 6 | 35 | 1.8 | 1.5 | 15.2 | 1.1 |  | BC with K-salt, K salts, sulfates |

| | | | | | | | | | | | |
|---|---|---|---|---|---|---|---|---|---|---|---|
| RF05_1 | 943 | Above | 154 | 6 | 75 | 3.4 | 0.6 | | 2.3 |  | BC mixed with NaSK salts, Ca bearing, Silicates |
| RF05_2 | 378 | Below | 106 | marine | 76 | 2.7 | 0.3 | 23.5 | 1.6 |  | NaSK salts, often mixed with BC |
| RF05_3 | 3247 | Above | 210 | 1 | 39 | 0.7 | | | 0.2 |  | organic, sulfate mixed with BC |
| RF06_1 | 2444 | Above | 248 | 2 | 17 | 1.9 | 9.5 | 19.3 | 1.4 |  | organic, organic with KS inclusions |
| RF06_2 | 2570 | Above | 173 | 2 | 53 | 5.9 | 1.2 | 38.9 | 4.0 |  | organic, organic/salt mixtures |
| RF07_1 | 1091 | Above | 121 | 6 | 53 | 7.1 | 5.9 | 15.4 | 3.1 |  | BC with K salts, OM, salts |
| RF07_2 | 159 | Below | 123 | marine | 52 | 7.0 | 0.6 | 20.8 | 3.6 |  | BC with silicate, Ca bearing, Al silicates |
| RF09 | 1307 | Above | 158 | 7 | 11 | 1.2 | | | 0.2 |  | organic with NSK salts, BC in fractal pattern |
| RF10 | 1986 | Above | 417 | 1 | 0 | | | | |  | externally mixed BC, and internally mixed with viscous sulfur coating. |
| RF11 | 3027 | Above | 190 | 2 | 5 | 0.8 | 0.2 | | 0.9 |  | ammonium sulphates, OM mixed with salts, externally mixed BC |
| RF13 | 1127 | Above | 118 | 4 | 13 | 1.9 | | | 0.3 |  | BC with K salt, OM, salts |

 **Table 5 Cl dominant aerosol on CLARIFY filters**

| Filter ID | Altitude (m) | Above/ Below cloud | CO (ppbv) | Time from fire (days) | Particle image & description | |
|-----------|--------------|--------------------|-----------|-----------------------|------------------------------|--|
| Gold_14 | 2845 | Above | 262 | 6 |  | N and Cl dominant spectra; uniform particles across filter |
| Gold_15 | 329 | Below | 158 | marine |  | C and Cl particles with small amount of K; uniform particles across filter |
| Gold_18 | 332 | Below | 119 | marine |  | Ca and Mg with Cl; uniform particles across filter |

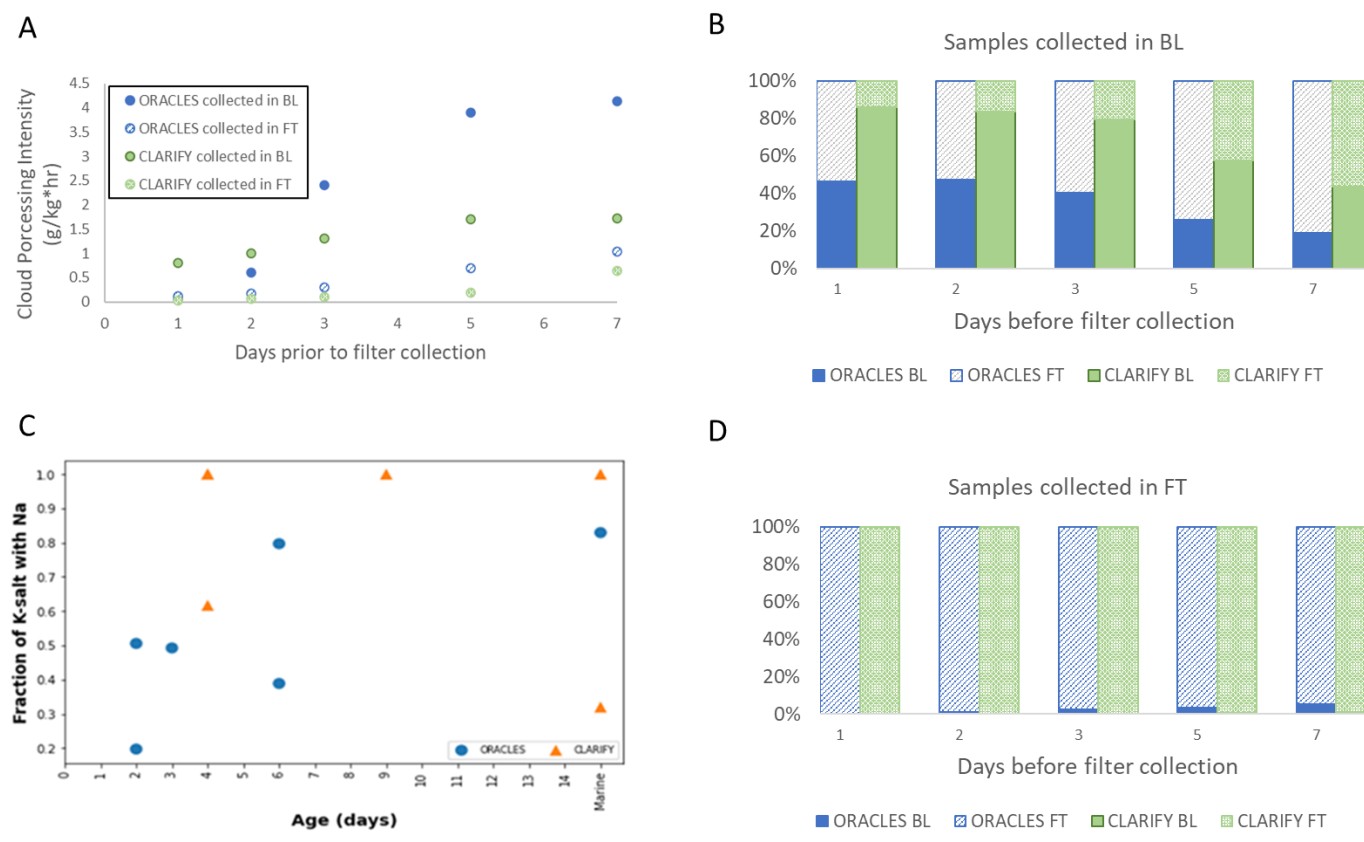

**Figure 7  Panel A) BL and FT ORACLES and CLARIFY cloud processing intensity  B) Time spent in BL and FT for samples collected in the BL in the days prior to filter collection  C) Fraction of K-salt, per filter, mixed with Na   D) Fractional time spent in BL and FT for samples collected in the FT in the days prior to filter collection**

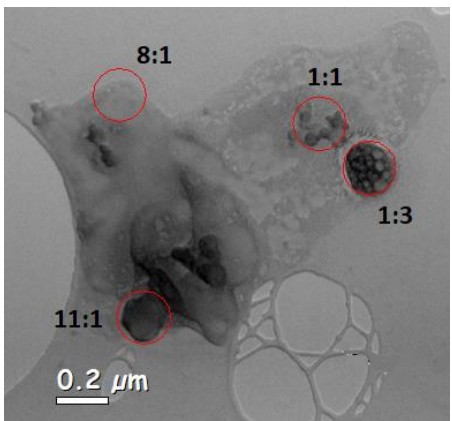

**Figure 8 Aqueous K-salt particle with varying levels of Na throughout particle, with Na:K weight ratio designated in red**

RF13_23
RH 55%
Tcloud_1_day: 6.94
LWC_mean 1 day: 0.038
13.14(K) 0.84(O)

RF5_Filter2_63
RH 30%
Tcloud_1_day: 8.42
LWC_mean 1 day: 0.012
1.48(S) 14.93(K) 5.18(O)

RF5_Filter1_47
RH 60%
Tcloud_1_day: 11.29
LWC_mean 1 day: 0.018
15.56(S) 25.01(K)
6.7(Na) 18.7(O)

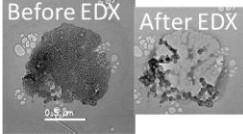

RF3_43
RH 60%
Tcloud_1_day: 18.08
LWC_mean 1 day: 0.052
5.87(S) 6.74(K) 5.24(O)
3.97(N)

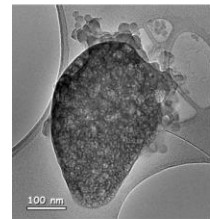
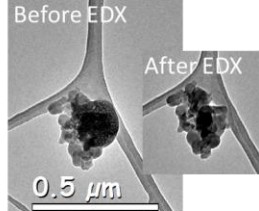
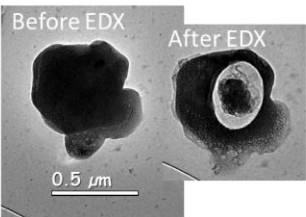

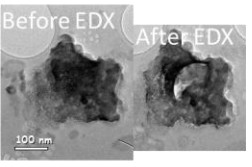

RF3_31
RH 60%
Tcloud_1_day: 18.08
LWC_mean 1 day: 0.052
2.67(S) 3.65(K) 2.91(O)

Time in cloud (1 day prior to filter collection)

1065

RF5Filter3_18
RH 60%
Tcloud_1_day: 0
LWC_mean 1 day: 0
6.5(S) 2.15(K)
0.41(Na) 1.66(O)

RF10_45
RH 75%
Tcloud_1_day: 2.25
LWC_mean 1 day: 0.006
5(S) 2.62(K) 2.24(O)

RF3_29
RH 55%
Tcloud_1_day: 9.35
LWC_mean 1 day: 0.052
4.27(S) 2.81(K) 1.41(O)

RF5_Filter1
RH 60%
Tcloud_1_day: 11.29
LWC_mean 1 day: 0.018
4.88(S) 2.48(K) 3.44(Na)
7.5(O)

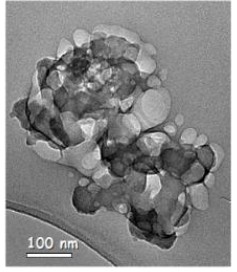
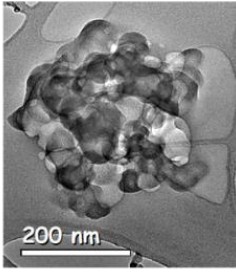
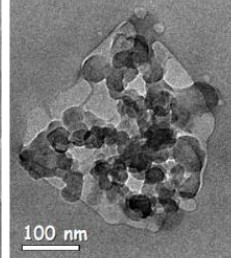
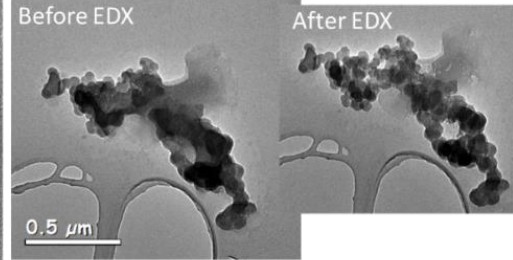

Time in cloud (1 day prior to filter collection)

**Figure 9 Potassium salt mixed with BC (top panel) and sulfur-organic and BC (bottom panel) as a function of time in cloud in the 24 hours prior to filter collection.**

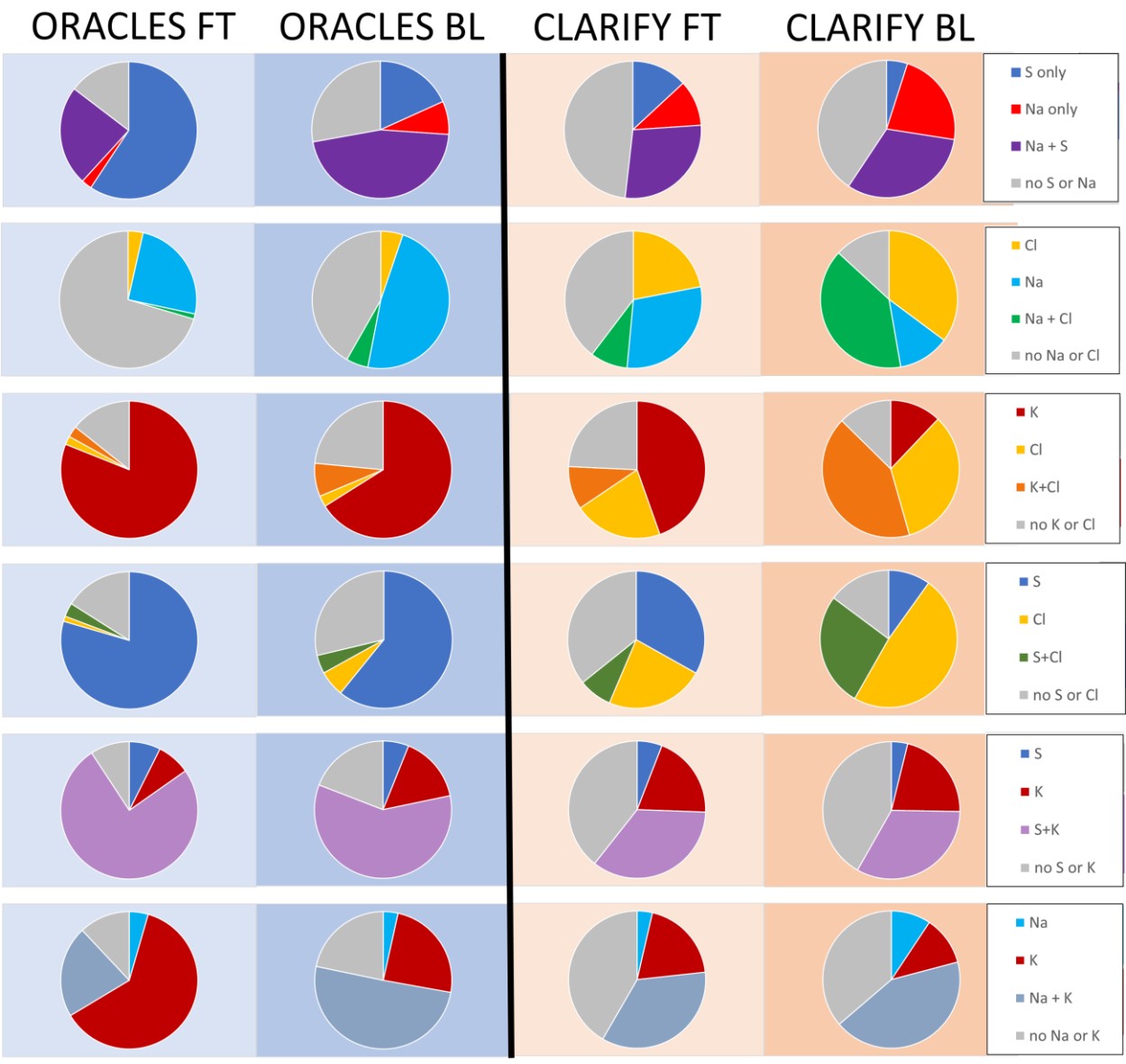

**Figure 10 Elemental mixing states for select elemental pairs for ORACLES (two left columns) and CLARIFY (two right columns), separated by filter collection in the free troposphere and boundary layer**

1070