# Peer review of "Biomass burning and marine aerosol processing over the southeast Atlantic Ocean: A TEM single particle analysis"

_Atmospheric Chemistry and Physics, 2021_

## Referee Comment (RC2)

General Comments

In their manuscript "Biomass burning and marine aerosol processing over the southeast Atlantic Ocean: A TEM single particle analysis," the authors Dang et al. present a series of analyses on aged biomass-burning and marine aerosols collected during the CLARIFY and ORACLES field campaigns sampling biomass-burning aerosol and biomass-burning impacted marine air masses originating from central Africa. The paper focuses on the transmission electron microscopy analysis of collected particles and the insights that can be gained from this analysis. The analysis is wide-ranging, the paper is overall well written, interesting results are presented, and the geographic area and emission timescale covered by the field-collected samples address a significant current measurement gap. However, there are several sections of the data presentation and discussion where I feel that the authors need to include more detail, either to support the conclusions that are drawn, to cover other possible explanations for the observations, or to address the results of prior work. I believe the manuscript will be suitable for publication in *Atmospheric Chemistry and Physics* after the questions and issues below are addressed.

1. The authors discuss the atmospheric processing of BBA in detail, but I'm not certain I agree with all aspects of the discussion and conclusions. The authors conclude with "Due to the considerable processing of organic aerosol and the noted effects of the MBL on BBA, it appears that aqueous processing, photolysis, and interaction with the MBL are the key drivers in physical and chemical properties such as mixing state and elemental composition of very aged BBA, rather than source" (page 16, lines 584-586).

The conclusion that fuel type is not a major factor in affecting composition and morphology of aged BBA appears to be based on observations that BBA becomes mixed with other inorganic salts through atmospheric and cloud processing and therefore the initial BBA composition is of lesser significance. This conclusion may be reasonable for BBA originating in the currently studied geographic region but does not seem generally applicable to BBA where interaction with marine air masses will not necessarily occur in the days following emission. Additionally, as the authors note, fuel type has a substantial influence on the relative and absolute prevalence of salt and chloride phases in BBA (e.g., Jahn et al. 2021; Goldberger et al. 2019; Liu et al. 2017; Levin et al. 2010) and salt phases will influence the ability of particles to uptake water and act as CCN (e.g., Gomez et al. 2018; Semeniuk et al. 2007; Pósfai et al. 2003). The present work suggests that cloud and/or aqueous processing can significantly impact BBA aging, so would the ability of fresh BBA to uptake water and act as CCN (which is strongly influenced by inorganic salt content and therefore fuel type) not be a potentially major driver in affecting how atmospheric aging proceeds?

The processing of organic aerosol is discussed in terms of organic ratios ($Org_{tot}$, BC, f44, f43) derived from AMS (and SP2?) measurements and from TEM observations on particle viscosity and volatility. The description of f44 and f43 in section 2.4 does not appear to be consistent with the cited work (Ng et al. 2011). Ng et al. stated that "m/z 44 is thought to be mostly due to acids… or esters" (not diacids or organic peroxides, page 5 line 190) and that "m/z 43 is predominantly due to non-acid oxygenates" (not acids, page 5 line 187). The authors mention derivation of f60 in section 2.4 but do not utilize this organic fraction during later analysis. Recent work (Hodshire et al. 2019) has highlighted the usage of the ratio f60:f44 to analyze BB plume aging in a number of studies: could such a comparison be useful in the present work as well? The authors seem to focus more on photolysis and fragmentation (in addition to aqueous processing) as ways that BBOA evolves during atmospheric aging and less on oxidation of

organics (in either the gas or particle phase), mentioning oxidation specifically only once (page 16, line 574). Oxidation increases the AMS oxidized mass fractions (f44 and f43), leads to oxidative fragmentation, and can change molecular photoabsorption cross sections. Photoxidation of laboratory-generated BBA has also been observed to alter BBOA volatilization behavior (Jahn et al. 2021) and physical properties (Jahl et al. 2021). I think the authors should address in more detail the role that oxidation may play in altering BBOA, in addition to photolysis, evaporation/condensation, and aqueous processing.

2. Can the authors provide more detail on sample collection? Was any size segregation performed during particle sampling? From the TEM images in the manuscript, it looks like many particles are in the submicron range; is this characteristic of all the particles that are included in the tabulated data and statistics? How long were individual filter sample collection times? Is it possible that particle morphologies or compositions were altered during collection, and if so could this affect aspects of the discussion on particle mixing state? For example: could particle impaction alter particle morphology due to, e.g., the spreading of organic material around other particle constituents or the lacey carbon framework; or could inorganic material undergo chemical reactions following collection as the aircraft passes into new airmasses?

3. I think the authors need to include more detail when describing the composition of different particle populations, specifically what's shown in Table 3 and Figure 11. Table 3 lists average ratios of Na:Cl and Na:S across CLARIFY filters but it's difficult to tell how much heterogeneity exists within the filter populations as a whole. Relatedly, Figure 11 illustrates the extent to which S, Na, K, and Cl are co-located within the ORACLE and CLARIFY samples but doesn't communicate anything about the elemental ratios, which makes it difficult to draw conclusions on particle composition, mixing state, and population-level compositional heterogeneity. I suggest the authors construct a series of ternary diagrams (either for the main text or SI) to more clearly show the ratios of these elements within particles, which will better allow the reader to understand and visualize the composition, mixing state, and heterogeneity of each filter's particle population. I think this would also make it easier for a reader to follow the discussion in section 3.0—Results and evaluate the conclusions in section 4—Conclusions.

Specific comments

Page 3, lines 84-85: I realize that this summary of the conclusions of Posfai et al (2003) is modeled after what's stated in the abstract of that work, but I don't believe that this is the most complete way to represent the conclusions of that work. Posfai et al. wrote "Even if the carbonaceous part of organic particles were water insoluble, the inorganic K-salt inclusions should make the mixed particle hydrophilic. Thus, organic particles with inorganic inclusions are likely responsible for the high cloud-nucleating potential of biomass smoke particles." Posfai et al. discussed the high degree of mixing between inorganics and both organics and soot, and in the abstract emphasized that observations on internal mixing informed their hypothesis on the cloud nucleus constituents of smoke. Thus, I think the authors of the present study should also emphasize that inorganic material was theorized to be important for BBA hygroscopic behavior in Posfai et al. Related to this, I would like to suggest another reference relevant for this discussion: the related work of Semeniuk et al. (2007), who analyzed the hygroscopic behavior of SAFARI BBA through TEM and concluded that inorganic particle components were the main driver behind particle water uptake and hygroscopic behavior.

Page 3, line 110: "very" is used to describe the degree of aging here and elsewhere; please be more quantitative when describing the degree of aging, either highlighting time since emission, time in cloud, and/or estimated radical exposure levels (or the ranges of these quantities, if more than one sample is being discussed).

Page 7, lines 242-244: what time period does each data point in Figure 2 (right) represent? What CO values would be considered typical of BB-influenced air vs relatively pristine marine air?

Page 7, line 249: I'm not sure what "groups" refers to in this sentence: individual particles? Different samples? Please clarify.

Page 8, lines 266-275: the observations and discussion on tar balls here is interesting. Can the authors comment on any potential reasons (differences in fuel or fire conditions, differences in other parameters measured during sampling, etc.) for why tar balls may have been observed in these two fresh samples but not others? Additionally, can the authors comment on any potential removal or transformation mechanisms?

Page 8, lines 285-288: do the authors intend to describe the mechanism by which BC is coated with inorganics? Prior work (Jahn et al. 2020; Li et al. 2003; Gaudichet et al. 1995) has described the formation and prevalence of inorganic phases in BBA as resulting (in part) from the volatilization-deposition of inorganic elements present within the biomass fuel, which can theoretically deposit to any particles generated in the BB plume.

Page 8, lines 288-289: was BC mixed with both organics and salt a mixing state that was observed at all?

Page 9, line 304: Prior electron microscopy work has indicated that there can be significant deviation from the nominal Na:Cl atomic ratio in sea spray aerosol, which is ~0.86:1 (not 1:1) and which also varies as a function of particle size (Krueger et al. 2003, see Figure 7). I suggest that the authors potentially consider additional metrics or ratios to use to examine the aging of sea spray aerosol, or at least discuss the natural variability that can be present.

Page 9, line 305: please provide a reference for the sea spray aerosol processing timeline. Is this timeline mostly universal for sea spray aerosol or does it vary by region or with influencing air masses?

Page 9, line 306: please provide citation(s) for the statement on $SO_2$ oxidation, as there are several mechanisms by which this can occur.

Page 9, lines 307-308: Cl would be volatilized as a gaseous molecule, not in its atomic form. Cl gases can also partition to existing Na-free aerosol and do not necessarily need to be involved in the process by which new particles form.

Page 9, line 308-309: please provide citations for the statement on determining SSA aging, as (I assume) this methodology has been described and used in prior work.

Page 9, line 320: I don't know how significantly this will affect the overall sea spray aging discussion, but I think it's worth acknowledging that prior work has observed significant variation (<1-13%) in the atomic % of S in fresh sea spray particles (Ault et al. 2013).

Page 9, line 324: specify the Na:Cl ratio on Gold 23.

Page 9, line 335: explain the rationale behind the 120 ppbv CO cutoff: is this based on other measurements from these field campaigns or prior literature?

Page 10, lines 345-346: are the high amounts of NOx based on measurements during the current campaigns or prior literature? Please specify and provide citations.

Page 10, lines 347-348: could the BC-NaNO₃ particle originate from biomass burning rather than sea spray? It is difficult to tell the size of the Na region(s) in Figure 6e. Please also specify in the text here that the proper TEM image is Figure 6e.

Pages 10-11: I have several questions related to filters Gold 14, 15, and 18 and the discussion here.

- Gold 8 was noted to have several Cl-rich particles and was described as having other markers for mixing between the boundary layer and free troposphere; could the Cl-rich particles on Gold 8 be related to those on Gold 14, 15, or 18 in any way?
- The composition of particles on Gold 14 (N and Cl) suggests that these may be composed of NH4Cl, and the discussion of the HNO3-NH3-NH4NO3 system seems to imply that the authors consider this possibility; do the authors think it's likely that the observed N-Cl particles are NH4Cl? If so, please state this directly.
- Based on the EDX data, the identity of the Cl⁻ counterion appears to be unclear in most of the Gold 15 particles; is this correct? Do the authors have any hypotheses regarding the Cl phase? Is it possible this is NH4Cl but the N signal is relatively weaker compared to Gold 14 and is therefore not visible?
- The composition of particles on Gold 18 (Mg/Ca and Cl) seems like it could suggest particles of a marine origin, as particles of similar composition (relatively enriched in Mg or Ca) and morphology appear to have been observed in prior work (Kirpes et al. 2018; Prather et al. 2013). Is the elemental composition of these particles homogeneous across each whole particle? If these particles originate from sea spray without additional processing, would that change the discussion in this section?
- Gold 18 has CO levels below 120 ppbv listed in Table 4 (119 ppbv), does this mean that this sample was not influenced at all by BB air as would be suggested based on the previously mentioned 120 ppbv cutoff? 120 vs 119 ppbv is obviously a small difference, but it illustrates the difficulties of using a sharp cutoff value.
- The authors propose that new particle formation occurred before/during flight C042, but this does not appear to be a period where new particle formation was discussed in the work of Wu et al. (2020) cited earlier in this section. Can the authors provide similar evidence (SMPS size distributions) for the assertion in the present work as was used in Wu et al. (2020)? Looking at Figure 7 it appears to me that very little time was spent in the altitude range 2250-2800 m and from Table 1 it appears that there is no AMS or SP2 data for this time period either; given the lack of data, is it reasonable to speculate on the composition and formation mechanisms of these particles? Or, are there alternative explanations to the one given in the manuscript?
- I disagree with the characterization of ammonium nitrate as semivolatile: ammonia and nitrate (as nitric acid) are volatile/semivolatile, but ammonium nitrate salt is not semivolatile.
- With the reported RH measurements, the authors seem to be implying that (at least some) aerosols are expected to be deliquesced; if this is the case, please state so directly. Prior work (Jahn et al. 2021; Semeniuk et al. 2007) has also observed BBA to take up water in this RH range.

- The work of Gunthe et al. (2021) concluded that high levels of ammonia were necessary to drive HCl partitioning to the aerosol phase. Are there any measurements during this or similar time periods that point towards what could potentially drive HCl to condense and react? I realize this may be beyond the scope of the current work, but can thermodynamic modelling (e.g., Pye et al. 2020) during this or similar periods be used to constrain the likelihood and conditions under which HCl could partition to the aerosol phase (e.g., how much HCl would need to condense to generate the observed particles, and is this amount reasonable given ambient sources and concentrations)?
- The inorganic/crustal elements observed on filters Gold 14, 15, and 18 (Si, K, Ca, Mg) have all been observed in BBA during prior work (Li et al. 2003); if the authors believe these signatures are from SSA and not BBA please make that clear.
- The authors posit (page 11, lines 389-391) that condensation of Cl-species gives rise to the particles observed on filters Gold 14, 15, and 18. Prior work examining sea spray aerosol suggests that sea spray composition is heterogeneous and can vary episodically depending on the aerosol generation mechanism and other conditions (Ault et al. 2013; Prather et al. 2013; Krueger et al. 2003). Is there a possibility that some of the particles on Gold 14, 15, and 18 are primary?

Page 12, line 420: my understanding of cloud microphysics is somewhat limited; is cloud processing intensity a generally used metric? If so, please include a citation, and if not, please explain the rationale. My understanding of droplet coalescence is that the overall size distribution is relevant for predicting coalescent collisions and that clouds with high aerosol loading can have high cloud water content but a low rate of coalescence.

Page 12, line 432: please provide a citation (e.g., a recent review article) that provides some background on cloud droplet coalescence processes.

Page 13, Figure 10/discussion before section 3.4: over what area is the elemental wt% measured in each particle? If possible, please note the area over which EDX was performed, as in Figure 9. Do the EDX wt% values listed include all non-C elements? For the particles shown in the top panel, do the authors have before/after EDX data to inform what phases volatilized from the particle during analysis? The authors write that these are K-phases, but it also looks like crystalline material remains after EDX on RF5_Filter1_47. Can the authors clarify what they mean by "degradation of the K-salt during cloud processing"? Is this chemical reaction to form mixed K-phases, some degree of amorphization originating from hydration/dissolution and recrystallization, or something else? Some salt phases (some nitrates and sulfates, for example) have also been observed to be unstable upon electron beam exposure following exposure to water vapor and/or acid gases without aqueous processing (e.g., Jahn et al. 2021; Hoffman et al. 2004), offering an alternative explanation for the behavior observed here.

Pages 13-14, section 3.4: I found the discussion in this section a little difficult to follow. The paragraphs list a lot of percentages that are difficult to mentally keep track of and I ended up rereading this section a couple times. I would consider reorganizing some of this material to start a paragraph/section with a conclusion statement and then go through the numbers that support it; for example, starting the paragraph at page 13, line 476 with the idea that BBA is more diluted on CLARIFY than ORACLES filters and then explaining the numbers that support this.

Page 14, line 504: consider emphasizing that aqueous formation pathways for sulfate in cloud water are generally predicted to be faster than-gas phase formation pathways.

Page 14, line 512: In a preceding paragraph the authors suggested a marine influence for the Cl in CLARIFY aerosol, and in the preceding section (3.2) and later in this paragraph the authors discuss the mixing/coalescence of sea spray and BB particles and the fact that sea spray aerosol may contain some amount of K. Therefore, I'm wondering why co-located K/S+Cl implies Cl condensation to BBA in CLARIFY aerosol rather than mixing of sea spray and BBA or simply the presence of sea spray aerosol? Or is the condensation of Cl implied solely for the filters described in Table 4?

Page 15, lines 528-529: I don't say this to lessen the relevance and importance of the present work, but Posfai et al. (2003) did observe a large degree of internal mixing between BC and salts.

Page 15, lines 539-540: consider rephrasing this sentence, as it implies (to me) that CLARIFY/ORACLES weren't focused on BBA and that SAFARI didn't attempt to examine the noted particle constituents (which they did).

Page 15, line 547 and page 16, line 558: consider combining these paragraphs, as they appear to discuss the same phenomenon.

**Technical Comments**

References to Dobracki et al. (2021), Sedlacek et al. (2021), and Che et al. (2021) are included in numerous locations throughout the manuscript but are not included in the list of references. Even if these manuscripts are not published, a bibliographic entry is needed for each with the author list, working title, and preparation status.

The word "collocate" is used in various places in the manuscript; I believe the authors intend to use the word co-locate (alternatively colocate), as collocate is a separate word

Page 2, line 53: "BBA aerosol" is redundant

Page 4, line 125: include the product ID# for the holey (lacey?) TEM grids and nuclepore filters

Page 5, line 206 (and potentially elsewhere): "above cloud samples" → "above-cloud samples"

Page 10, line 369: add that Wu et al. (2020) is also based on the CLARIFY campaign to emphasize the relevance to the discussion here

Page 10, line 372: add the CPC to the instrumentation listed in the Methods section

Page 12, line 417: there's an extra "spent"

Page 13, line 459: "about" → "amount"

General Table and Figure comments: please try to increase the size of axis labels, legends, and in-figure labels, as these were overall difficult to read. I also urge the authors to use a, b, c… labelling for figure panels to enable clearer references to specific panels within the manuscript. As many of the TEM images in tables are too small to see clearly, consider including larger images in a supplementary file.

Page 25, Figure 5: three numbers are present in the ratio above the left-most TEM image but the order and identity of the third number isn't listed

Page 26, Table 3 caption: "forum" → "from"

Page 30, Figure 11: The ORACLES and CLARIFY labels within the figure are switched from the caption

**References**

Ault, A.P., Moffet, R.C., Baltrusaitis, J., Collins, D.B., Ruppel, M.J., Cuadra-Rodriguez, L.A., Zhao, D., Guasco, T.L., Ebben, C.J., Geiger, F.M., Bertram, T.H., Prather, K. a., and Grassian, V.H. (2013). Size-Dependent Changes in Sea Spray Aerosol Composition and Properties with Different Seawater Conditions. *Environ. Sci. Technol.* 47 (11):5603–5612. doi:10.1021/es400416g.

Gaudichet, A., Echalar, F., Chatenet, B., Quisefit, J.P., Malingre, G., Cachier, H., Buat-Menard, P., Artaxo, P., Maenhaut, W., Radioactivitds, F., and Gent, R. (1995). Trace elements in tropical African savanna biomass burning aerosols. *J. Atmos. Chem.* 22 (1):19–39. doi:10.1007/BF00708179.

Goldberger, L.A., Jahl, L.G., Thornton, J.A., and Sullivan, R.C. (2019). N2O5 reactive uptake kinetics and chlorine activation on authentic biomass-burning aerosol. *Environ. Sci. Process. Impacts* 21 (10):1684–1698. doi:10.1039/C9EM00330D.

Gomez, S.L., Carrico, C.M., Allen, C., Lam, J., Dabli, S., Sullivan, A.P., Aiken, A.C., Rahn, T., Romonosky, D., Chylek, P., Sevanto, S., and Dubey, M.K. (2018). Southwestern U.S. Biomass Burning Smoke Hygroscopicity: The Role of Plant Phenology, Chemical Composition, and Combustion Properties. *J. Geophys. Res. Atmos.* 123 (10):5416–5432. doi:10.1029/2017JD028162.

Hodshire, A.L., Akherati, A., Alvarado, M.J., Brown-Steiner, B., Jathar, S.H., Jimenez, J.L., Kreidenweis, S.M., Lonsdale, C.R., Onasch, T.B., Ortega, A.M., and Pierce, J.R. (2019). Aging Effects on Biomass Burning Aerosol Mass and Composition: A Critical Review of Field and Laboratory Studies. *Environ. Sci. Technol.* 53 (17):10007–10022. doi:10.1021/acs.est.9b02588.

Hoffman, R.C., Laskin, A., and Finlayson-Pitts, B.J. (2004). Sodium nitrate particles: physical and chemical properties during hydration and dehydration, and implications for aged sea salt aerosols. *J. Aerosol Sci.* 35 (7):869–887. doi:10.1016/j.jaerosci.2004.02.003.

Jahl, L.G., Brubaker, T.A., Polen, M.J., Jahn, L.G., Cain, K.P., Bowers, B.B., Fahy, W.D., Graves, S., and Sullivan, R.C. (2021). Atmospheric aging enhances the ice nucleation ability of biomass-burning aerosol. *Sci. Adv.* 7 (9):eabd3440. doi:10.1126/sciadv.abd3440.

Jahn, L.G., Jahl, L.G., Bowers, B.B., and Sullivan, R.C. (2021). Morphology of Organic Carbon Coatings on Biomass-Burning Particles and Their Role in Reactive Gas Uptake. *ACS Earth Sp. Chem.* 5 (9):2184–2195. doi:10.1021/acsearthspacechem.1c00237.

Jahn, L.G., Polen, M.J., Jahl, L.G., Brubaker, T.A., Somers, J., and Sullivan, R.C. (2020). Biomass combustion produces ice-active minerals in biomass-burning aerosol and bottom ash. *Proc. Natl. Acad. Sci.* 117 (36):21928–21937. doi:10.1073/pnas.1922128117.

Kirpes, R.M., Bondy, A.L., Bonanno, D., Moffet, R.C., Wang, B., Laskin, A., Ault, A.P., and Pratt, K.A. (2018). Secondary sulfate is internally mixed with sea spray aerosol and organic aerosol in the winter Arctic. *Atmos. Chem. Phys.* 18 (6):3937–3949. doi:10.5194/acp-18-3937-2018.

Krueger, B.J., Grassian, V.H., Iedema, M.J., Cowin, J.P., and Laskin, A. (2003). Probing Heterogeneous Chemistry of Individual Atmospheric Particles Using Scanning Electron Microscopy and Energy-Dispersive X-ray Analysis. *Anal. Chem.* 75 (19):5170–5179. doi:10.1021/ac034455t.

Levin, E.J.T., McMeeking, G.R., Carrico, C.M., Mack, L.E., Kreidenweis, S.M., Wold, C.E., Moosmüller, H., Arnott, W.P., Hao, W.M., Collett, J.L., and Malm, W.C. (2010). Biomass burning smoke aerosol

properties measured during Fire Laboratory at Missoula Experiments (FLAME). *J. Geophys. Res.* 115 (D18):D18210. doi:10.1029/2009JD013601.

Li, J., Pósfai, M., Hobbs, P. V, and Buseck, P.R. (2003). Individual aerosol particles from biomass burning in southern Africa: 2, Compositions and aging of inorganic particles. *J. Geophys. Res. Atmos.* 108 (D13). doi:10.1029/2002JD002310.

Liu, L., Kong, S., Zhang, Y., Wang, Y., Xu, L., Yan, Q., Lingaswamy, A.P., Shi, Z., Lv, S., Niu, H., Shao, L., Hu, M., Zhang, D., Chen, J., Zhang, X., and Li, W. (2017). Morphology, composition, and mixing state of primary particles from combustion sources - Crop residue, wood, and solid waste. *Sci. Rep.* 7 (1):1–15. doi:10.1038/s41598-017-05357-2.

Ng, N.L., Canagaratna, M.R., Jimenez, J.L., Chhabra, P.S., Seinfeld, J.H., and Worsnop, D.R. (2011). Changes in organic aerosol composition with aging inferred from aerosol mass spectra. *Atmos. Chem. Phys.* 11 (13):6465–6474. doi:10.5194/acp-11-6465-2011.

Pósfai, M., Simonics, R., Li, J., Hobbs, P. V, and Buseck, P.R. (2003). Individual aerosol particles from biomass burning in southern Africa: 1. Compositions and size distributions of carbonaceous particles. *J. Geophys. Res. Atmos.* 108 (D13). doi:10.1029/2002JD002291.

Prather, K.A., Bertram, T.H., Grassian, V.H., Deane, G.B., Stokes, M.D., DeMott, P.J., Aluwihare, L.I., Palenik, B.P., Azam, F., Seinfeld, J.H., Moffet, R.C., Molina, M.J., Cappa, C.D., Geiger, F.M., Roberts, G.C., Russell, L.M., Ault, A.P., Baltrusaitis, J., Collins, D.B., Corrigan, C.E., Cuadra-Rodriguez, L.A., Ebben, C.J., Forestieri, S.D., Guasco, T.L., Hersey, S.P., Kim, M.J., Lambert, W.F., Modini, R.L., Mui, W., Pedler, B.E., Ruppel, M.J., Ryder, O.S., Schoepp, N.G., Sullivan, R.C., and Zhao, D. (2013). Bringing the ocean into the laboratory to probe the chemical complexity of sea spray aerosol. *Proc. Natl. Acad. Sci. U. S. A.* 110 (19):7550–7555. doi:10.1073/pnas.1300262110.

Pye, H.O.T., Nenes, A., Alexander, B., Ault, A.P., Barth, M.C., Clegg, S.L., Collett, J.L., Fahey, K.M., Hennigan, C.J., Herrmann, H., Kanakidou, M., Kelly, J.T., Ku, I.T., Faye McNeill, V., Riemer, N., Schaefer, T., Shi, G., Tilgner, A., Walker, J.T., Wang, T., Weber, R., Xing, J., Zaveri, R.A., and Zuend, A. (2020). The acidity of atmospheric particles and clouds. *Atmos. Chem. Phys.*

Semeniuk, T.A., Wise, M.E., Martin, S.T., Russell, L.M., and Buseck, P.R. (2007). Hygroscopic behavior of aerosol particles from biomass fires using environmental transmission electron microscopy. *J. Atmos. Chem.* 56 (3):259–273. doi:10.1007/s10874-006-9055-5.

---

## Author Comment (AC1)

*We thank the reviewers for their detailed and helpful comments. We have added discussions based on reviewers' comments, adjusted multiple figures, and added ternary diagrams in the Supplementary Material. We have also adjusted the structure of the paper. Our responses are below in the italicized font, and the track-changed manuscript and Supplement is attached. Line numbers refer to the revised draft.*

**Reviewer 1**

This manuscript presents an interesting data set on the composition of aerosol sampled during ORACLES and CLARIFY downwind of Africa over the Atlantic Ocean during burning season in 2017 and 2018. Single particle elemental analysis conducted with TEM-EDX provides information about the mixing state (internal versus external), shape, viscosity, and volatility of the collected particles that can provide insight into photochemical, aqueous and heterogenous processing during transport which is not available from techniques measuring bulk aerosol.

The authors suggest that the new data presented in this paper provide important complement to previously published AMS data from the 2 campaigns that will address 3 key questions: 1) do the aerosols in the 2 campaigns differ as a function of the age of the smoke plumes, 2) what are the differences between aerosol in FT and MBL, and 3) what processes caused any differences found while investigating the first 2 questions? Authors also state that determining the mixing state of the aerosol is a motivating question, but I assert that the mixing state is information that may contribute to answering the other questions rather than being of high intrinsic interest by itself. The evolution of smoke advected over the ocean, and how that evolution may be modified by mixing with seasalt aerosol, as well as how seasalt aerosol may be modified by mixing with smoke, have climatic relevance so I was looking forward to hearing the answers to the 3 questions the authors stated they were addressing.Unfortunately, bulk of the paper presents the single particle composition in a large number of ways that are not very well connected to each other, and rarely clearly connected back to the motivating questions. The fact that the conclusions of this manuscript include nothing related to the fact that CLARIFY sampled older smoke than ORACLES, and almost nothing about differences between aerosol in the FT and MBL (just that there was less depletion of Cl in fresh seasalt aerosol in the MBL compared to in the FT) is illustrative. Also, more than a few interesting (sometimes puzzling) observations are pointed out almost in passing, with little attempt at understanding what they may suggest about aerosol processing.

*We have removed the mixing state as a motivating question and have tried to more clearly address the* **motivating** *questions, as most sections refer to both the different campaigns and age and BL/FT. We have also rewritten most of the conclusion.*In the following I work through the paper noting sections that could/should be expanded to perhaps develop one or more compelling arguments. Along the way I will also note sections or statements that are not as clear as they might be.

In 3ʳᵈ paragraph of the introduction the summary of prior work on the extent of coating on BC during CLARIFY (very thick) compared to ORACLES (less thick with some evidence for loss of coating with age) is interesting since the smoke sampled in CLARIFY was significantly older and might therefore have less coating. I assumed this was setting up a major line of investigation based on the TEM-EDX results, but really never found it.

*Regarding coating, we found a significant amount of inorganic coating in both campaigns and now emphasize this in the conclusion and line 350. Less organic observed with the TEM in more aged aerosol (CLARIFY) is also consistent with Dobracki et al.'s and Sedlacek III et al.'s findings of loss of organic with plume age, although our TEM results are qualified by loss of volatile organic in the chamber. By less organic found in CLARIFY, this also extends to less coating being present and also less thick than CLARIFY. We discuss this in the OA section (lines 266, 305) and in the conclusion lines 735-755*

In second paragraph of section 2.3 it might be helpful to say something about where most of the fires were. Alternatively, modifications to Fig 1 or supplemental Fig 1 might be a way to convey this information (more on that later).

*Added that fires were in southern and central Africa (lines 242)*

Second paragraph of section 2.4, were there SP2 instruments on both aircraft? If not, which had one and which did not?

*Yes, there were SP2 instruments on both aircraft. We have added this (lines 218-220).*

Lines 202-203 state that Table 1 includes BC mass/total PM1 but I do not find that. Rather it includes BC mass and the number of BC particles/cm^3.

*Thank you, have fixed this (line 230)*

Second paragraph of section 3.1 introducing Fig 1. Lot of questions about the trajectories: where were the fires, how far above the fire(s) did given parcel pass (i.e., many trajectories quite high, hence may not have entrained much smoke)? It might be easier for the reader to figure some of this out if there were 4 panels, separating BL and FT samples, with fire hot spots included on the maps.

*Figure 1 was introduced for general orientation purposes and doesn't rely on landcover or trajectories to identify if a plume was sampled. Included in text (lines 244-246): Detailed flight information is included in overview papers including whether the flight passed through a plume (Redemann et al 2021). Ancillary data (CO data) will show whether a plume was sampled, with models (Redemann et al. 2021) showing that plumes are often above the cloud.*

Line 216 Table 1 shows a filter collected 8/30/17 and 2 on 9/30/18 so "except for two" should be except for three.

*Thank you, we have changed this (line 243)*

Section 3.2.1 The finding that the organic aerosol in CLARIFY was more volatile (hence lost more quickly when hit by TEM beam) is interesting, and a noteworthy finding. But it is not clear to me that the AMS f44 fully supports this. While there are a few ORACLES samples with high f44 most of the samples in both campaigns cluster from 0.18 to.24 (per right panel in Fig 2). Please explain how

the combination of f43 and f44 is, or may not be, consistent with your inference based on single particle analysis.

> *Add (lines 283-286): Most of the variation in filters sampled is in the ORACLES points with higher f44 than the CLARIFY data. As f44 is an indicator of low OOA fraction but not high volatility fraction, the higher ORACLEs points with regard to f44 is consistent with TEM findings of lower volatility organic on ORACLES filters. The f43 spread is similar to differences in instrument baselines and therefore should not be over interpreted.*

The third paragraph in this section is also interesting, using TEM images to assess viscosity and volatility. But there are very few particles shown in Fig 3. Would be much stronger with a numerical summary of how many particles in ORACLES looked like the round one on left compared to the one in the middle, etc.

> *Added (lines 296): Included that more than 80% of particles in ORACLES are rounded/viscous as shown in the left and center images in Figure 3, top panel.*

The loss of tarballs with age is an important finding.

Section 3.2.2 I find the first sentence to be a little confusing, by including internally and externally mixed variants of K-salts. If you counted a particle that was mostly OA, with a little K (or BC with small K crystal attached) as OA (or BC) instead of K would the number fraction of K-salts be much smaller?

> *Added lines (326-328): Only K-salts that looked solid were counted in this number. So, if a particle was OA with K, but without K-salt inclusion, this would not be counted as a K-salt. If a particle was BC with a K-crystal attached, that would be a BC-K salt internally mixed particle*

More important point. Table 2 shows pretty big difference between BL and FT in all three columns, but the sign of all differences is opposite between the two campaigns. Given motivating question 2 (and 3) nearly requires that the authors at least try to explain this.

> *Added lines (347-353) Table 2 shows a difference between BL and FT in all columns, with the sign of the differences being different in the two campaigns. It should be noted that of the three ORACLES filters collected in the BL, two have marine backtrajectories, so BB organic may be underrepresented here. For CLARIFY, cloud processing may remove the more hygroscopic BC containing particles as these are activated and removed by precipitation, and hence the organic/BC ratio is high relative to the FT, but this does not work for ORACLES. The main finding here is that BC with inorganic, as analyzed by TEM, is the most prevalent BC mixing state.*

Section 3.2.3 I am not convinced that it is a good decision to ignore ORACLES in this section. Kind of seems the point of entire paper is to compare and contrast the 2 campaigns, in the framework of looking at old versus very old smoke (and how both kinds of smoke interact with seasalt).

*We agree with this. We have added Table 4 to analyze ORACLES filters in the same manner as Table 3's CLARIFY and discuss in section 3.2.3*

Line 301. The atomic ratio of Na:Cl in SS is 0.84. Seasalt is not halite.

*Change this, thank you, to 0.86:1 based on additional input from Reviewer 2.*

Displacement of Cl from seasalt by acids is well established (based on many studies), so need not be emphasized so much here. The Cl rich particles are probably new, hence more interesting.

Lines 348-349 Not convinced that Na in the FT (mixed with BBA) requires mixing BBA into MBL and then modified aerosol back out. Could just mix seasalt into FT and have

*Good point, changed sentence to include sea salt mixing into FT (line 435)*

On the other hand, the Gold1 filter does support BBA mixing into the MBL (but is this really surprising?)

359-366 As noted above, the Cl rich particle are interesting. But are they important (e.g., what impact on radiative forcing, good CCN)?

*We have not found much information on these Cl rich particles in terms of atmospheric importance, but would hypothesize that they are good CCN based on ability to uptake water (lines 459-460)*

Section 3.3 last paragraph? Any evidence that the change in the response of K-salts to the electron beam translates to an atmospherically relevant change (like the decreasing viscosity of the sulfur/organic particles in the same samples)

*Based on Reviewer 2's input, we have specified that degradation of K-salts may be due to amorphization, dissolution and recrystallization of the salt during processing. (lines 560-562)*

*We hypothesize that a more amorphous structure may allow for more water uptake and that the salts may be better CCNs or more susceptible for further processing, but have not included in this text because pretty speculative.*

Section 3.4 end of second paragraph. Dilution of particles sourced from fires is not unexpected as transport distance increases. Does the similar fraction of the S/K particles in FT and BL during CLARIFY suggest more mixing between the FT and BL (compared to ORACLES). Is there a meteorological reason this might happen (like more convection, as suggested elsewhere based on transition in cloud field)

*Added that this may be due to entrainment/detrainment into the lower FT (lines 592-594).*

lines 502-505 Case for more S bearing particles in the FT during ORACLES being due to cloud processing is not really supported by Fig 8a and lines 420-425 where it is clear that it was in the MBL

that ORACLES samples had more cloud influence.

*We have removed "FT"*

Lines 507-519  This is a jumpy paragraph that is hard to glean anything from.  Not clear that colocation of K, Cl and S requires gas phase HCl condensing on BBA as seasalt contains all 3 of these elements.

*Yes this paragraph is very jumpy; we have deleted the paragraph and reworked the entire section 3.4.*

Section 4 (Conclusions)  See opening comment regarding lack of connection to motivating questions.

*We have reworked the conclusion to try to more clearly address differences in campaigns due to aging, as well as potential processing and SSA-BBA interaction and coating/organic*

Lines 528-530  Given the loss of OA in the TEM, the large fraction of internally mixed K and BC is artificially inflated, perhaps by a lot in CLARIFY samples.

*This is true and we have added this.*

Line 539-540  First sentence of next paragraph has no real point

*This is true; we have deleted the sentence.*

Lines 553-554  see comment above on lines 507-519

*We have mostly rewritten this paragraph.*

**Reviewer 2**

General Comments

In their manuscript "Biomass burning and marine aerosol processing over the southeast Atlantic Ocean: A TEM single particle analysis," the authors Dang et al. present a series of analyses on aged biomass-burning and marine aerosols collected during the CLARIFY and ORACLES field campaigns sampling biomass-burning aerosol and biomass-burning impacted marine air masses originating from central Africa. The paper focuses on the transmission electron microscopy analysis of collected particles and the insights that can be gained from this analysis. The analysis is wide-ranging, the paper is overall well written, interesting results are presented, and the geographic area and emission timescale covered by the field-collected samples address a significant current measurement gap. However, there are several sections of the data presentation and discussion where I feel that the authors need to include more

detail, either to support the conclusions that are drawn, to cover other possible explanations for the observations, or to address the results of prior work. I believe the manuscript will be suitable for publication in *Atmospheric Chemistry and Physics* after the questions and issues below are addressed.

1. The authors discuss the atmospheric processing of BBA in detail, but I'm not certain I agree with all aspects of the discussion and conclusions. The authors conclude with "Due to the considerable processing of organic aerosol and the noted effects of the MBL on BBA, it appears that aqueous processing, photolysis, and interaction with the MBL are the key drivers in physical and chemical properties such as mixing state and elemental composition of very aged BBA, rather than source" (page 16, lines 584-586).

The conclusion that fuel type is not a major factor in affecting composition and morphology of aged BBA appears to be based on observations that BBA becomes mixed with other inorganic salts through atmospheric and cloud processing and therefore the initial BBA composition is of lesser significance. This conclusion may be reasonable for BBA originating in the currently studied geographic region but does not seem generally applicable to BBA where interaction with marine air masses will not necessarily occur in the days following emission. Additionally, as the authors note, fuel type has a substantial influence on the relative and absolute prevalence of salt and chloride phases in BBA (e.g., Jahn et al. 2021; Goldberger et al. 2019; Liu et al. 2017; Levin et al. 2010) and salt phases will influence the ability of particles to uptake water and act as CCN (e.g., Gomez et al. 2018; Semeniuk et al. 2007; Pósfai et al. 2003). The present work suggests that cloud and/or aqueous processing can significantly impact BBA aging, so would the ability of fresh BBA to uptake water and act as CCN (which is strongly influenced by inorganic salt content and therefore fuel type) not be a potentially major driver in affecting how atmospheric aging proceeds?

> *This is a great point, thank you. We have incorporated the potential importance of inorganic salt content in fresh BBA mixing into the conclusion (765-771) as well as in the abstract. Further we have added that gas phase oxidation of NOx leading to formation of NO$_3$ is a significant pathway for further addition of inorganic salt.*

The processing of organic aerosol is discussed in terms of organic ratios (Org$_{tot}$, BC, f44, f43) derived from AMS (and SP2?) measurements and from TEM observations on particle viscosity and volatility. The description of f44 and f43 in section 2.4 does not appear to be consistent with the cited work (Ng et al. 2011). Ng et al. stated that "m/z 44 is thought to be mostly due to acids… or esters" (not diacids or organic peroxides, page 5 line 190) and that "m/z 43 is predominantly due to non-acid oxygenates" (not acids, page 5 line 187). The authors mention derivation of f60 in section 2.4 but do not utilize this organic fraction during later analysis. Recent work (Hodshire et al. 2019) has highlighted the usage of the ratio f60:f44 to analyze BB plume aging in a number of studies: could such a comparison be useful in the present work as well? The authors seem to focus more on photolysis and fragmentation (in addition to aqueous processing) as ways that BBOA evolves during atmospheric aging and less on oxidation of organics (in either the gas or particle phase), mentioning oxidation specifically only once (page 16, line 574). Oxidation increases the AMS oxidized mass fractions (f44 and f43), leads to oxidative fragmentation, and can change molecular photoabsorption cross sections. Photoxidation of laboratory-generated BBA has also been observed to alter BBOA volatilization behavior (Jahn et al. 2021) and physical properties (Jahl et al. 2021). I think the authors should address in more detail the role that oxidation may play in altering BBOA, in addition to photolysis, evaporation/condensation, and aqueous processing.

*We have changed to the m/z 43 and m/z44 descriptions to account for acid/esters and non-acid oxygenates as stated by Ng et al. (2011). (lines 211 and 214)*

*Based on conversations with colleagues, using f60 is challenging since it is close to background based on m60 values in clean and BB plume conditions. We have removed the f60 reference (line 208)*

*We have included more information on oxidation throughout the manuscript (lines 306-311,755-760).*

2. Can the authors provide more detail on sample collection? Was any size segregation performed during particle sampling? From the TEM images in the manuscript, it looks like many particles are in the submicron range; is this characteristic of all the particles that are included in the tabulated data and statistics? How long were individual filter sample collection times? Is it possible that particle morphologies or compositions were altered during collection, and if so could this affect aspects of the discussion on particle mixing state? For example: could particle impaction alter particle morphology due to, e.g., the spreading of organic material around other particle constituents or the lacey carbon framework; or could inorganic material undergo chemical reactions following collection as the aircraft passes into new airmasses?

> *We have included text to say (lines 153-164): Size segregation was not performed during particle sampling. Most observed particles are in the submicron range. It is possible that morphologies or compositions were altered during collection, as in other aerosol TEM studies. For example compositions of hydrate sulfates have been suggested to change in the TEM chamber or during processing (Buseck and Pósfai 1999),with acidic particles containing more water spreading more on a TEM grid than neutral species. Andreae et al. (1986) suggest that CaSO4 observed on filters without sea salt ions in the marine atmosphere could be from breakup up sea salt particles containing a gypsum crystallite. A sodium chloride core and magnesium chloride coating has been suggested to be due to efflorescence of a particle after collection (Ault et al. 2013). Posfai et al. suggest that an interesting crystalline rod morphology may be due to water loss within the TEM chamber. Generally, the particles we observed were separated from other particles on the filter and so agglomeration and aggregation did not influence organic mixing with adjacent particles. Samples were collected, on average, for approximately ten minutes and in dry conditions, which may limit any chemical reactions the particles are subject to as the aircraft passes into new air masses.*

3. I think the authors need to include more detail when describing the composition of different particle populations, specifically what's shown in Table 3 and Figure 11. Table 3 lists average ratios of Na:Cl and Na:S across CLARIFY filters but it's difficult to tell how much heterogeneity exists within the filter populations as a whole. Relatedly, Figure 11 illustrates the extent to which S, Na, K, and Cl are co-located within the ORACLE and CLARIFY samples but doesn't communicate anything about the elemental ratios, which makes it difficult to draw conclusions on particle composition, mixing state, and population-level compositional heterogeneity. I suggest the authors construct a series of ternary diagrams (either for the main text or SI) to more clearly show the ratios of these elements within particles, which will better allow the reader to understand and visualize the composition, mixing state, and heterogeneity of each filter's particle population. I think this would also make it easier for a reader to follow the discussion in section 3.0—Results and evaluate the conclusions in section 4—Conclusions.

*Na-S-K and Na-S-Cl ternary diagrams are included in the Supplementary Material. They show, broadly, more Na for below cloud samples in ORACLES and higher Na for both above below and above cloud samples in CLARIFY than ORACLES.*

Specific comments

Page 3, lines 84-85: I realize that this summary of the conclusions of Posfai et al (2003) is modeled after what's stated in the abstract of that work, but I don't believe that this is the most complete way to represent the conclusions of that work. Posfai et al. wrote "Even if the carbonaceous part of organic particles were water insoluble, the inorganic K-salt inclusions should make the mixed particle hydrophilic. Thus, organic particles with inorganic inclusions are likely responsible for the high cloud-nucleating potential of biomass smoke particles." Posfai et al. discussed the high degree of mixing between inorganics and both organics and soot, and in the abstract emphasized that observations on internal mixing informed their hypothesis on the cloud nucleus constituents of smoke. Thus, I think the authors of the present study should also emphasize that inorganic material was theorized to be important for BBA hygroscopic behavior in Posfai et al. Related to this, I would like to suggest another reference relevant for this discussion: the related work of Semeniuk et al. (2007), who analyzed the hygroscopic behavior of SAFARI BBA through TEM and concluded that inorganic particle components were the main driver behind particle water uptake and hygroscopic behavior.

*Included in lines 87-90: They determined that organic particles with inorganic inclusions likely contribute to the high cloud nucleating capability of biomass burning particles, and Semeniuk et al. (2007), using environmental TEM, found that the inorganic phases of SAFARI particles took up water while soot and tar balls did not; therefore they determined that the inorganic content of mixed organic-inorganic particles determined the hygroscopic properties of BBA.*

Page 3, line 110: "very" is used to describe the degree of aging here and elsewhere; please be more quantitative when describing the degree of aging, either highlighting time since emission, time in cloud, and/or estimated radical exposure levels (or the ranges of these quantities, if more than one sample is being discussed).

*We have changed/removed "very" here and in other instances throughout the paper and replaced it with the age in days.*

Page 7, lines 242-244: what time period does each data point in Figure 2 (right) represent? What CO values would be considered typical of BB-influenced air vs relatively pristine marine air?

*We have added (lines 275-278) ORACLES filters (triangles) are 2-7 days and CLARIFY (squares) are 4-15 days aged. A CO cutoff value of over 120 ppbv is used to denote BB-influenced air, based on overall campaign data and Figure 17 in Haywood et al. (2021), which shows the Ascension Island CO frequency distribution and that 120 is at the upper end of the Gaussian distribution of the clean air data.*

Page 7, line 249: I'm not sure what "groups" refers to in this sentence: individual particles? Different samples? Please clarify.

*Thank you, this was a confusing sentence. Changed to "particles." (line 290)*

Page 8, lines 266-275: the observations and discussion on tar balls here is interesting. Can the authors comment on any potential reasons (differences in fuel or fire conditions, differences in other parameters measured during sampling, etc.) for why tar balls may have been observed in these two fresh samples but not others? Additionally, can the authors comment on any potential removal or transformation mechanisms?

> *This suggests a removal process, potentially through deep precipitation near the coast, as they are advected west over the ocean.*

Page 8, lines 285-288: do the authors intend to describe the mechanism by which BC is coated with inorganics? Prior work (Jahn et al. 2020; Li et al. 2003; Gaudichet et al. 1995) has described the formation and prevalence of inorganic phases in BBA as resulting (in part) from the volatilization-deposition of inorganic elements present within the biomass fuel, which can theoretically deposit to any particles generated in the BB plume.

> *We have added (lines 331-336): Inorganic salts in BBA can result from volatiles from the burning source depositing inorganics onto particles in the BB plume (Jahn et al., 2020; Li et al., 2003; Gaudichet et al., 1995). Different salts will indicate different processes; K-salts will form due to evaporation of potassium in the fire and subsequent near field condensation onto the BC; while this will occur with some S and N as well, co-emitted $SO_2$ and $NO_2$ can oxidize and condense and lead to additional coating in the far field.*

Page 8, lines 288-289: was BC mixed with both organics and salt a mixing state that was observed at all?

> *We did not observe this.*

Page 9, line 304: Prior electron microscopy work has indicated that there can be significant deviation from the nominal Na:Cl atomic ratio in sea spray aerosol, which is ~0.86:1 (not 1:1) and which also varies as a function of particle size (Krueger et al. 2003, see Figure 7). I suggest that the authors potentially consider additional metrics or ratios to use to examine the aging of sea spray aerosol, or at least discuss the natural variability that can be present.

> *We added that natural variability can be present, with Krueger et al. (2003) finding that Cl/Na atomic ratio increases with particle diameter. (lines 378-380)*

Page 9, line 305: please provide a reference for the sea spray aerosol processing timeline. Is this timeline mostly universal for sea spray aerosol or does it vary by region or with influencing air masses?

> *We have added lines (377-381 ) However, Na and Cl in the sea salt aerosol rarely are in a 0.86:1 ratio as would be expected from freshly emitted SSA, indicating that the particles have been processed. Natural variability can be present, with Krueger et al.(2003) finding that Cl/Na atomic ratio in sea salts increases with particle diameter. The aging timescale of sea salt also varies depending on the production of $NO_2$ and $SO_2$ and its conversion rate to $H_2SO_4$ and $HNO_3$ since these acids displace the Cl, and these rates will vary by location.*

Page 9, line 306: please provide citation(s) for the statement on $SO_2$ oxidation, as there are several

mechanisms by which this can occur.

*We have provided Sievering et al. 1991 and Miller et al.(1987) as references. (lines 383-384)*

Page 9, lines 307-308: Cl would be volatilized as a gaseous molecule, not in its atomic form. Cl gases can also partition to existing Na-free aerosol and do not necessarily need to be involved in the process by which new particles form.

*Removed sentence*

Page 9, line 308-309: please provide citations for the statement on determining SSA aging, as (I assume) this methodology has been described and used in prior work.

*Cl/Na ratios for relative aging have been use for example in Kirpes et al. (2018), Hand et al. (2010) and Young et al. (2016). (lines 385-386)*

Page 9, line 320: I don't know how significantly this will affect the overall sea spray aging discussion, but I think it's worth acknowledging that prior work has observed significant variation (<1-13%) in the atomic % of S in fresh sea spray particles (Ault et al. 2013).

*Have added this reference (line 391)*

Page 9, line 324: specify the Na:Cl ratio on Gold 23.

*Specified, as 20.2*

Page 9, line 335: explain the rationale behind the 120 ppbv CO cutoff: is this based on other measurements from these field campaigns or prior literature?

*As mentioned in a previous answer, based on overall campaign data; Figure 17 in Haywood et al. (2021) showing the Ascension Island CO frequency distribution and 120 is at the upper end of the Gaussian distribution of the clean air data.*

Page 10, lines 345-346: are the high amounts of NOx based on measurements during the current campaigns or prior literature? Please specify and provide citations.

*Prior literature, referenced Jin et al. (2021) for high levels of NOx in BB plumes (line 433)*

Page 10, lines 347-348: could the BC-NaNO$_3$ particle originate from biomass burning rather than sea spray? It is difficult to tell the size of the Na region(s) in Figure 6e. Please also specify in the text here that the proper TEM image is Figure 6e.

*We have specified that the TEM image is Figure 6e and also that possible this may be sea spray mixing into the FT with BC. (lines 436-437)*

Pages 10-11: I have several questions related to filters Gold 14, 15, and 18 and the discussion here.

- Gold 8 was noted to have several Cl-rich particles and was described as having other markers for

mixing between the boundary layer and free troposphere; could the Cl-rich particles on Gold 8 be related to those on Gold 14, 15, or 18 in any way?

*Yes, the Cl rich particles in Gold 8 have a similar morphology to those in Gold 14,15,18 so could be related. However 30% of the particles on Gold 8 has the presence of Na, so is different from the Gold14, 15, 18 filters, which did not have any Na present. (lines 409-411)*

- The composition of particles on Gold 14 (N and Cl) suggests that these may be composed of NH4Cl, and the discussion of the HNO3-NH3-NH4NO3 system seems to imply that the authors consider this possibility; do the authors think it's likely that the observed N-Cl particles are NH4Cl? If so, please state this directly.

*The presence of ammonium in the FT and Cl in the samples, as well as strong Cl and N peaks in the EDX spectra suggest that these particles may be NH4Cl. (lines 479-480)*

- Based on the EDX data, the identity of the Cl- counterion appears to be unclear in most of the Gold 15 particles; is this correct? Do the authors have any hypotheses regarding the Cl phase? Is it possible this is NH4Cl but the N signal is relatively weaker compared to Gold 14 and is therefore not visible?

*The lack of an observable counterion in the EDX spectra of Gold 15 particles may be due to a weaker N signal than in Gold 14, or potentially Cl dispersed within a sol-gel network. (lines 489-490)*

- The composition of particles on Gold 18 (Mg/Ca and Cl) seems like it could suggest particles of a marine origin, as particles of similar composition (relatively enriched in Mg or Ca) and morphology appear to have been observed in prior work (Kirpes et al. 2018; Prather et al. 2013). Is the elemental composition of these particles homogeneous across each whole particle? If these particles originate from sea spray without additional processing, would that change the discussion in this section?

*Possibly, this is interesting. We did not perform EDX mapping but the composition appears homogenous across the whole particle, and we did not observe any distinct coatings.*

*Prather et al. (2013) observed that long chain bioorganic species as well as Ca and Mg form stable collapsed structures as sol-gels, and potentially particles from Gold 15 and Gold 18 may be sol-gel with Ca and Mg dispersed within a sol gel structure. (lines 487-488)*

- Gold 18 has CO levels below 120 ppbv listed in Table 4 (119 ppbv), does this mean that this sample was not influenced at all by BB air as would be suggested based on the previously mentioned 120 ppbv cutoff? 120 vs 119 ppbv is obviously a small difference, but it illustrates the difficulties of using a sharp cutoff value.

*We agree the cutoff is rather sharp and 119 may well be BB influenced air.*

- The authors propose that new particle formation occurred before/during flight C042, but this does not appear to be a period where new particle formation was discussed in the work of Wu et al. (2020) cited earlier in this section. Can the authors provide similar evidence (SMPS size distributions) for the assertion in the present work as was used in Wu et al. (2020)? Looking at Figure 7 it appears to me that

very little time was spent in the altitude range 2250-2800 m and from Table 1 it appears that there is no AMS or SP2 data for this time period either; given the lack of data, is it reasonable to speculate on the composition and formation mechanisms of these particles? Or, are there alternative explanations to the one given in the manuscript?

*We have changed the graph  as the sampling time indicates a shorter time period than was indicated in the Gold 14 figure.  It is now F2 in the Supplementary Material.*
*We do not have SMPS size distributions for this time period, and CN data shows around 1437 particles/cm³, which although high, is not definitive.  .  As such we will also include primary SSA aerosol as a potential mechanism.*

- I disagree with the characterization of ammonium nitrate as semivolatile: ammonia and nitrate (as nitric acid) are volatile/semivolatile, but ammonium nitrate salt is not semivolatile.

*Makes sense, thank you.  We have removed semivolatile from the text.*

- With the reported RH measurements, the authors seem to be implying that (at least some) aerosols are expected to be deliquesced; if this is the case, please state so directly. Prior work (Jahn et al. 2021; Semeniuk et al. 2007) has also observed BBA to take up water in this RH range.

*Relative humidity at filter collection times according to backtrajectories is in the 60-70% range, which would facilitate aerosol deliquescnce and the subsequent uptake of HCl.  (lines 473-475)*

- The work of Gunthe et al. (2021) concluded that high levels of ammonia were necessary to drive HCl partitioning to the aerosol phase. Are there any measurements during this or similar time periods that point towards what could potentially drive HCl to condense and react? I realize this may be beyond the scope of the current work, but can thermodynamic modelling (e.g., Pye et al. 2020) during this or similar periods be used to constrain the likelihood and conditions under which HCl could partition to the aerosol phase (e.g., how much HCl would need to condense to generate the observed particles, and is this amount reasonable given ambient sources and concentrations)?

*There are no ammonia measurements from the campaign, but AMS ammonium measurements from other filters from the campaign range from 0.1 to 3 ug/cm⁻³.  Unfortunately, we do not have AMS NH4 data for the Gold 14, 15 and 18 filters.  We think that thermodynamic modelling is a great idea regarding HCl partitioning.*

- The inorganic/crustal elements observed on filters Gold 14, 15, and 18 (Si, K, Ca, Mg) have all been observed in BBA during prior work (Li et al. 2003); if the authors believe these signatures are from SSA and not BBA please make that clear.

*The presence of the elements Si, K, Ca and Mg are hypothesized to be from SSA rather than biomass burning based on the unique morphology and composition of the particles on these filters. (lines 459-461)*

- The authors posit (page 11, lines 389-391) that condensation of Cl-species gives rise to the particles observed on filters Gold 14, 15, and 18. Prior work examining sea spray aerosol suggests that sea spray

composition is heterogeneous and can vary episodically depending on the aerosol generation mechanism and other conditions (Ault et al. 2013; Prather et al. 2013; Krueger et al. 2003). Is there a possibility that some of the particles on Gold 14, 15, and 18 are primary?

> *Yes, potentially they are primary, we were assuming that they were secondary based on the lack of Na present. We have added: The particles could also originate from sea spray without additional processing, and as primary particles may be sol gel structures of bioorganics and as observed in Prather et al. (2013). (lines 501-502)*

Page 12, line 420: my understanding of cloud microphysics is somewhat limited; is cloud processing intensity a generally used metric? If so, please include a citation, and if not, please explain the rationale. My understanding of droplet coalescence is that the overall size distribution is relevant for predicting coalescent collisions and that clouds with high aerosol loading can have high cloud water content but a low rate of coalescence.

> *Cloud processing intensity is a novel metric described in detail in Che et al. (submitted). (lines 525-526)*

Page 12, line 432: please provide a citation (e.g., a recent review article) that provides some background on cloud droplet coalescence processes.

> *Grabowski et and Wang (2013) has been added. (line 536)*

Page 13, Figure 10/discussion before section 3.4: over what area is the elemental wt% measured in each particle? If possible, please note the area over which EDX was performed, as in Figure 9. Do the EDX wt% values listed include all non-C elements? For the particles shown in the top panel, do the authors have before/after EDX data to inform what phases volatilized from the particle during analysis? The authors write that these are K-phases, but it also looks like crystalline material remains after EDX on RF5_Filter1_47. Can the authors clarify what they mean by "degradation of the K-salt during cloud processing"? Is this chemical reaction to form mixed K-phases, some degree of amorphization originating from hydration/dissolution and recrystallization, or something else? Some salt phases (some nitrates and sulfates, for example) have also been observed to be unstable upon electron beam exposure following exposure to water vapor and/or acid gases without aqueous processing (e.g., Jahn et al. 2021; Hoffman et al. 2004), offering an alternative explanation for the behavior observed here.

> *We do not have the specific area over which EDX was performed, other than areas where there is a clear hole in the particle- that is where the EDX beam was, for example RF5_Filter1_47. The beam was centered on a large and central part of each particle for this analysis. The EDX wt% includes all non-C elements. While it would have been very useful to have before/after EDX data to inform the phases of what happened after beam exposure, we do not have this data. For example RF5_Filter1_47, the electron beam created a hole in the particle, so the crystalline part which remains was outside of the electron beam. We have clarified degradation of K-salt to include amorphization from hydration, dissolution and recrystallization, as suggested, and have also included the references regarding water vapor and acid gases. (lines 562-564)*

Pages 13-14, section 3.4: I found the discussion in this section a little difficult to follow. The paragraphs list a lot of percentages that are difficult to mentally keep track of and I ended up rereading this section a couple times. I would consider reorganizing some of this material to start a paragraph/section with a conclusion statement and then go through the numbers that support it; for example, starting the paragraph at page 13, line 476 with the idea that BBA is more diluted on CLARIFY than ORACLES filters and then explaining the numbers that support this.

*We have reorganized and rewritten most of section 3.4.*

Page 14, line 504: consider emphasizing that aqueous formation pathways for sulfate in cloud water are generally predicted to be faster than-gas phase formation pathways.

*We have emphasized this in the text (lines 597-598)*

Page 14, line 512: In a preceding paragraph the authors suggested a marine influence for the Cl in CLARIFY aerosol, and in the preceding section (3.2) and later in this paragraph the authors discuss the mixing/coalescence of sea spray and BB particles and the fact that sea spray aerosol may contain some amount of K. Therefore, I'm wondering why co-located K/S+Cl implies Cl condensation to BBA in CLARIFY aerosol rather than mixing of sea spray and BBA or simply the presence of sea spray aerosol? Or is the condensation of Cl implied solely for the filters described in Table 4?

*We have reworked this section to include the possibility of primary sea spray aerosol.*

Page 15, lines 528-529: I don't say this to lessen the relevance and importance of the present work, but Posfai et al. (2003) did observe a large degree of internal mixing between BC and salts.

*Changed this to note that Posfai et al. (2003) as well as our work show a large amount of mixing between black carbon and salts.  (lines 671-672)*

Page 15, lines 539-540: consider rephrasing this sentence, as it implies (to me) that CLARIFY/ORACLES weren't focused on BBA and that SAFARI didn't attempt to examine the noted particle constituents (which they did).

*Removed sentence*

Page 15, line 547 and page 16, line 558: consider combining these paragraphs, as they appear to discuss the same phenomenon.

*Paragraphs combined*

**Technical Comments**
References to Dobracki et al. (2021), Sedlacek et al. (2021), and Che et al. (2021) are included in numerous locations throughout the manuscript but are not included in the list of references. Even if these manuscripts are not published, a bibliographic entry is needed for each with the author list, working title, and preparation status.

*References have been added/updated for these papers*

The word "collocate" is used in various places in the manuscript; I believe the authors intend to use the word co-locate (alternatively colocate), as collocate is a separate word

*Thank you, we have changed this throughout the document.*

Page 2, line 53: "BBA aerosol" is redundant

*Fixed to remove "with BBA"*

Page 4, line 125: include the product ID# for the holey (lacey?) TEM grids and nuclepore filters

*Fixed Ted Pella, Inc, Product # 01881 and WhatmanTM WHA10417112*

Page 5, line 206 (and potentially elsewhere): "above cloud samples" ▯ "above-cloud samples"

*Thank you, we have fixed this.*

Page 10, line 369: add that Wu et al. (2020) is also based on the CLARIFY campaign to emphasize the relevance to the discussion here

*Fixed*

Page 10, line 372: add the CPC to the instrumentation listed in the Methods section

*Added this*

Page 12, line 417: there's an extra "spent"

*fixed*

Page 13, line 459: "about" ▯ "amount"

*Fixed*

General Table and Figure comments: please try to increase the size of axis labels, legends, and in-figure labels, as these were overall difficult to read. I also urge the authors to use a, b, c… labelling for figure panels to enable clearer references to specific panels within the manuscript. As many of the TEM images in tables are too small to see clearly, consider including larger images in a supplementary file.
Page 25, Figure 5: three numbers are present in the ratio above the left-most TEM image but the order and identity of the third number isn't listed

*It's a ratio of 1.4:1 for page 25, Figure 5.  Increased size/ changed Figures 4, 7, 9, 10,*

Page 26, Table 3 caption: "forum" ▯ "from"

*Fixed*

Page 30, Figure 11: The ORACLES and CLARIFY labels within the figure are switched from the caption

*Fixed*

**Figure S4 Na-S-Cl ternary diagrams for A) ORACLES below cloud B) ORACLES above cloud C) CLARIFY below cloud and D) CLARIFY above cloud. Note that the lack of particles in A) and B) are due to the majority of particles having no Cl as well as most Cl-containing particles not containing Na or S.**

[revised manuscript text omitted]

**Figure S2  Particle count and altitude for Cl-dominant filters Gold 14, Gold 1, and Gold 18 during filter exposure times**

[Figure]

**Figure S3** Na-S-K ternary diagrams for A) ORACLES below cloud B) ORACLES above cloud C) CLARIFY below cloud and D) CLARIFY above cloud.

[Figure]

**Figure S4 Na-S-Cl ternary diagrams for A) ORACLES below cloud B) ORACLES above cloud C) CLARIFY below cloud and D) CLARIFY above cloud. Note that the dearth of particles in A) and B) are due to the majority of particles having no Cl as well as most Cl-containing particles not containing Na or S.**

---

## Referee Report (RR1)

I believe the paper is much improved in terms of the overall presentation, discussion and analysis of results, and the summary of the conclusions and significance. I have only a couple minor comments. Line numbers below refer to the new manuscript without tracked changes.

- Page 17, lines 597-600: "This is important as it suggests that the salts formed in the fire via evaporation and recondensation drive the mixing of the carbon aerosol as the secondary inorganic condenses, and that the organic fraction is separate. This is consistent with findings regarding emissions of BC and K-salts and other salts in the flaming phase of a fire, while organic emissions occur during the pyrolysis or smoldering phases (Haslett et al., 2018)."
  The discussion quoted above is interesting and provides useful insight on BBA mixing state. However, I would point out that the work of (Haslett et al. 2018) and related work of (Fawaz et al. 2021) utilized uniform sections of woody biomass combusted under tightly controlled conditions, which can differ from field conditions where temperature gradients may exist within the fire and fuel can be highly variable (in terms of leafy vs woody and/or wet vs dry). This can lead to some degree of spatial or temporal variability in fire emissions and may impact the very near-source mixing of fresh BBA. I don't believe that the quoted passage needs to be altered, but I do suggest the authors consider whether to add a caveat to this discussion.
- Page 17, lines 606-609: "While the SAFARI campaign and other recent biomass burning campaigns found tar balls, our TEM analysis did not find tar balls other than on filters RF10 and RF11, which were aged for approximately 1 and 2 days, respectively. This finding implies a reduction in tar balls in aged African BB plumes."
  Would this paragraph be an appropriate place to discuss potential implications of tar ball loss with aging, for example regarding the evolution of optical properties of African BB plumes? While a major area of tar ball research has been their optical properties, I realize there are still many unknowns regarding the specifics of tar ball optical properties as well as transformation or loss processes, however the direct TEM observations in the present work give the authors a unique position to comment.

Fawaz, M., Avery, A., Onasch, T.B., Williams, L.R., and Bond, T.C. (2021). Technical note: Pyrolysis principles explain time-resolved organic aerosol release from biomass burning. *Atmos. Chem. Phys.* 21 (20):15605–15618. doi:10.5194/acp-21-15605-2021.

Haslett, S.L., Thomas, J.C., Morgan, W.T., Hadden, R., Liu, D., Allan, J.D., Williams, P.I., Keita, S., Liousse, C., and Coe, H. (2018). Highly controlled, reproducible measurements of aerosol emissions from combustion of a common African biofuel source. *Atmos. Chem. Phys.* 18 (1):385–403. doi:10.5194/acp-18-385-2018.

---

## Author Response (AR3)

We thank both reviewers for their time and comments and address their comments below. Line numbers refer to the track changed version on the manuscript.

Referee 1

In my review of the original version of this manuscript I noted that the wide variety of particles that were sampled and described was kind of interesting, but there was no coherent narrative establishing progressive evolution of aged BBA nor how the changes during aging might be optically or biogeochemically important. This revised version is perhaps even less coherent. Nearly every particle type now has multiple hypotheses advanced in explanation, with nearly no attempt to develop arguments favoring one theory over the other.

It is now not clear whether the authors are confident that the TEM EDX results can distinguish between SSA and BBA after processing, nor whether the unusual Cl rich particles are secondary or primary. The lack of tarballs in the older smoke sampled in these studies compared to abundant tarballs in SAFARI remains interesting, but suggesting "a removal process, potentially through deep precipitation" that selectively scrubs tarballs but not the rest of BBA is not at all helpful.

Thank you for your comments. We address the evolution of BBA, TEM-EDX, describe optical property implications, Cl-rich particles, and tar balls below.

We think there is a narrative of evolution of BBA described in the paper, and summarize here. A major theme of the paper is the interaction between sea salt aerosol with BB plumes and biomass burning aerosol with marine air. Our findings are that BB aerosol are affected by the marine boundary layer and cloud processing as they age, and that organic becomes increasingly volatile with aging. Na and/or Cl can be mixed with the black carbon, potassium salts and organic from biomass burning plumes through aqueous processing or secondary processing. BB plumes and the higher levels of $NO_x$ and $SO_x$ can affect sea salt aging through more rapid Cl depletion. There is evidence of BL and FT top-of-cloud mixing as the BB plume advects west. TEM EDX results can distinguish between SSA and BBA, primarily because SSA will have a predominance of Na and/or Cl, and BBA aerosol will have more K in the form of potassium salts and/or the presence of BC. TEM is useful in that in can detect small changes in individual particle composition and processing on a single particle level, for example increased Na mixing with BC, so while there can seem to be a merging of particle types- we still can distinguish source based on the main element and their prevalence in individual particles. Caveats throughout the paper (ie, Cl and Na elements have been found in biomass burning particles) are for context, but if a particle is predominantly Na or Cl then we deduct that it is from a marine source; further we use ancillary data such as CO, altitude at sampling and backtrajectories to inform our assessments.

The unusual Cl rich particles were found on filters Gold 14, 15, and 18. While Gold 14 was collected above-cloud and EDX spectra showed strong Cl and N peaks, Gold 15 was collected below-cloud with C, Cl and small amount of K and Si. Gold 18 had Ca and Mg mixed with the Cl. Different mechanisms were proposed for the three filters based on elemental composition. We cannot assess whether they are secondary or primary, given that these types of particles have not been often observed in field observations, but put forth different potential pathways for their formation such as HCl uptake onto

liquid particles or gas-to-aerosol partitioning, and Ca and Mg distributed in a sol-gel particle for the Cl particles containing Ca and Mg based on prior studies (lines 426 to 462).

As for climate implications, radiative effects due to inorganic mixing with BC and changes in in hygroscopicity are noted in lines 604-607.  Further we note tar ball effects on models in lines 620 to 627: "Tar ball incorporation in BB models (Jacobson 2014) have been hindered due to lack of data, as tar balls can only be definitively detected with time-intensive single particle electron microscopy.  While other work shows that TBs are a significant fraction of BB aerosol, with some showing that tar balls outnumber BC by a factor of 10 (Hand et al. 2005; China et al., 2013), our analysis shows a lack of tar balls in the aged BB plumes, consistent with Posfai et al. (2003) who also reported a dearth of tar balls in aged plumes as a puzzling phenomenon.  Since tar balls are a light-absorbing particle, the absorption from aged plumes is dominated by non- tar ball components like BC, brown carbon, dust; the absence of tar balls in these aged plumes can help constrain models on radiative forcing in the region."

As the processes for tar ball removal are not clear, we have changed the line to : "This suggests a removal process, and while there are many unknowns regarding loss processes for tar balls, precipitation near the coast or heterogeneous, photolytically-driven processes which may affect the solubility or volatility of tarballs as they are advected west over the ocean may contribute to their removal. Posfai et al. (2003) also reported a dearth of tarballs when sampling in the haze layers representing aged BB plumes, without a clear explanation for their absence." (lines 313-317)

**Referee 2**

 I believe the paper is much improved in terms of the overall presentation, discussion and analysis of results, and the summary of the conclusions and significance. I have only a couple minor comments. Line numbers below refer to the new manuscript without tracked changes.

Page 17, lines 597-600: "This is important as it suggests that the salts formed in the fire via evaporation and recondensation drive the mixing of the carbon aerosol as the secondary inorganic condenses, and that the organic fraction is separate. This is consistent with findings regarding emissions of BC and K-salts and other salts in the flaming phase of a fire, while organic emissions occur during the pyrolysis or smoldering phases (Haslett et al., 2018)."

The discussion quoted above is interesting and provides useful insight on BBA mixing state. However, I would point out that the work of (Haslett et al. 2018) and related work of (Fawaz et al. 2021) utilized uniform sections of woody biomass combusted under tightly controlled conditions, which can differ from field conditions where temperature gradients may exist within the fire and fuel can be highly variable (in terms of leafy vs woody and/or wet vs dry). This can lead to some degree of spatial or temporal variability in fire emissions and may impact the very near-source mixing of fresh BBA. I don't believe that the quoted passage needs to be altered, but I do suggest the authors consider whether to add a caveat to this discussion.

We have added a caveat that field conditions may also affect the near- source mixing for fresh BBA, aside from the smoldering and flaming phases.  (lines 609-611)

▢ Page 17, lines 606-609: "While the SAFARI campaign and other recent biomass burning campaigns found tar balls, our TEM analysis did not find tar balls other than on filters RF10 and RF11, which were aged for approximately 1 and 2 days, respectively. This finding implies a reduction in tar balls in aged African BB plumes."

Would this paragraph be an appropriate place to discuss potential implications of tar ball loss with aging, for example regarding the evolution of optical properties of African BB plumes? While a major area of tar ball research has been their optical properties, I realize there are still many unknowns regarding the specifics of tar ball optical properties as well as transformation or loss processes, however the direct TEM observations in the present work give the authors a unique position to comment.

Yes, thank, you, we discuss the implications in the reply to Referee 1.

Fawaz, M., Avery, A., Onasch, T.B., Williams, L.R., and Bond, T.C. (2021). Technical note: Pyrolysis principles explain time-resolved organic aerosol release from biomass burning. *Atmos. Chem. Phys.* 21 (20):15605–15618. doi:10.5194/acp-21-15605-2021.
Haslett, S.L., Thomas, J.C., Morgan, W.T., Hadden, R., Liu, D., Allan, J.D., Williams, P.I., Keita, S., Liousse, C., and Coe, H. (2018). Highly controlled, reproducible measurements of aerosol emissions from combustion of a common African biofuel source. *Atmos. Chem. Phys.* 18 (1):385–403. doi:10.5194/acp-18-385-2018.